# Global and Arctic climate sensitivity enhanced by changes in North Pacific heat flux

Summer Praetorius [1,2], Maria Rugenstein [3], Geeta Persad[2] & Ken Caldeira[2]

Arctic amplification is a consequence of surface albedo, cloud, and temperature feedbacks, as well as poleward oceanic and atmospheric heat transport. However, the relative impact of changes in sea surface temperature (SST) patterns and ocean heat flux sourced from different regions on Arctic temperatures are not well constrained. We modify ocean-to-atmosphere heat fluxes in the North Pacific and North Atlantic in a climate model to determine the sensitivity of Arctic temperatures to zonal heterogeneities in northern hemisphere SST patterns. Both positive and negative ocean heat flux perturbations from the North Pacific result in greater global and Arctic surface air temperature anomalies than equivalent magnitude perturbations from the North Atlantic; a response we primarily attribute to greater moisture flux from the subpolar extratropics to Arctic. Enhanced poleward latent heat and moisture transport drive sea-ice retreat and low-cloud formation in the Arctic, amplifying Arctic surface warming through the ice-albedo feedback and infrared warming effect of low clouds. Our results imply that global climate sensitivity may be dependent on patterns of ocean heat flux in the northern hemisphere.

[1] United States Geological Survey, Menlo Park, CA 94025, USA. [2] Department of Global Ecology, Carnegie Institution for Science, Stanford, CA 94305, USA. [3] Institute for Atmospheric and Climate Science, ETH Zurich, 8092 Zurich, Switzerland. Correspondence and requests for materials should be addressed to S.P. (email: spraetorius@usgs.gov)

The Arctic (66.6–90°N) is the region experiencing the most rapid increase in surface temperature in response to global warming[1–4]; it is projected to warm by about three times the global average in future greenhouse gas emission scenarios[4]. The amplified climate response in the Arctic makes this region particularly susceptible to tipping points in both physical and ecological systems in the near future, with loss of summer sea ice often argued to be the most imminent[5,6]. Arctic warming and sea-ice decline is progressing at rates faster than most models project[7], indicating that there are processes contributing to Arctic amplification that are not being fully captured in the current generation of global climate models.

The feedbacks underlying Arctic amplification include the ice-albedo feedback, lapse rate, and Plank feedbacks, which lead to less efficient re-radiation to space at high-latitudes, as well as cloud feedbacks[8–11]. However, the role of the subpolar extra-tropics in modulating these feedbacks is not well-understood. Observed warming of the mid-troposphere in the Arctic indicates a role for enhanced atmospheric meridional heat transport in recent Arctic warming[1,12,13], most likely linked to increasing tropical and extratropical ocean temperatures. However, it is unclear which ocean regions may dominate this enhanced heat transport, and therefore exert the greatest influence on Arctic temperature and sea-ice feedbacks.

Decadal-scale modes of variability such as the Pacific Decadal Oscillation and Atlantic Multidecadal Oscillation have been implicated in Arctic sea-ice variability[2,14,15], indicating patterns of ocean-atmospheric variability may affect Arctic climate differently. Record-breaking Arctic warming and sea-ice decline has ensued in recent years[16], paralleling anomalous ocean warming in the Northeast Pacific[17] (Fig. 1a, c). Furthermore, sea surface temperature (SST) pattern formation is predicted to be non-stationary in response to changes in radiative forcing, with the most pronounced increase in surface ocean temperatures projected to occur in the North Pacific in the coming decades to centuries (Fig. 1b); an effect attributed to the shallow mixed layer depth that allows fast equilibration with the atmosphere[18,19]. We refer to this amplified SST response in the North Pacific on decadal to centennial time scales in response to radiative forcing as the 'North Pacific Rapid Response' (as distinguished from slower responses that might occur after adjustment of large ice sheets and deep ocean properties). In contrast to the North Pacific Rapid Response, SSTs in the subpolar North Atlantic are predicted to remain lower than global averages in the coming century due to deeper convection and mixing with cool subsurface waters (Fig. 1b, e.g., ref. [20]).

Paleoclimate records reveal episodes of abrupt ocean warming in the mid-latitude North Pacific[21–23] and North Atlantic[24,25] during the last deglaciation that were coeval with exceptionally rapid warming events of ~15°C in Greenland that occurred within decades to centuries[26] (Fig. 1c). These warming events were also accompanied by enhanced meridional flow of Pacific moisture transport across Alaska[27], implying a strong coupling of extratropical northern hemisphere SST changes with major reorganizations of poleward moisture transport and rapid Arctic climate change. These past ocean warming events had similar rates of SST increase to those predicted for the coming century under the RCP 8.5 scenario (~2–5°C/100 yr)[4,22], and were associated with the crossing of major ecological thresholds across the northern hemisphere, including widespread North Pacific ocean hypoxia[22,28] and Holarctic megafauna extinctions[29].

Given the observed coupling between abrupt Arctic temperature fluctuations and rapid SST changes in both the North Atlantic and North Pacific in the past, and the projected asymmetries in SST patterns in the near future, it is important to disentangle the relative impacts of changes in ocean heat flux sourced from each basin on Arctic temperature and sea ice extent to identify the teleconnections to which the Arctic may be most sensitive. Understanding such remote controls on Arctic climate will not only help to identify mechanisms that may lead to non-linear Arctic climate response in relation to changing SST patterns, but may help to reveal the primary sources of model bias that lead to underestimation of Arctic warming and sea-ice decline.

We imposed a series of ocean-to-atmosphere heat flux perturbations to the North Pacific and North Atlantic (30–60°N) in a slab-ocean configuration of the National Center for Atmospheric Research's Community Earth System Model (CESM) version 1.0.4[30] (Supplementary Figure 1, Table 1)[31] to assess the relative climatic impacts of changes in ocean heat flux from each basin on Arctic surface temperature and climate feedbacks. We impose both positive (into the atmosphere) (NP-Warm, NA-Warm) and negative (into the ocean) heat fluxes in each basin (NP-Cool, NA-Cool), aggregating to a global average of ±1 W/m². We also impose heat fluxes in both basins to compensate each other, such that the only change is in effective zonal heat transport and the net global energy perturbation is zero (NP-Cool/NA-Warm, NP-Warm/NA-Cool, i.e., the seesaw simulations).

## Results

**Seesaw heat flux perturbations.** In the seesaw simulations, the temperature response of the Arctic is dominated by the sign of the heat flux imposed on the North Pacific, such that an increase in North Pacific heat flux and SST results in an increase in Arctic temperature, despite an equivalent heat flux into the ocean imposed on the North Atlantic (Fig. 2). The NP-Cool/NA-Warm simulation results in Arctic cooling (−1.88 ± 0.14°C) and sea-ice advance (+4.07 ± 0.21% of total ice fraction), as well as a global cooling of −0.39 ± 0.03°C relative to the control simulation, despite an initial perturbation of 0 W/m² in the global mean. The NP-Warm/NA-Cool simulation results in an overall more heterogeneous response, with a mean Arctic warming of +0.34 ± 0.22°C and a mean global temperature change of −0.03 ± 0.03°C relative to the control simulation. The difference in the magnitude of the Arctic climate impacts between these simulations in part reflects the non-linear response to positive and negative ocean heat fluxes. Negative perturbations result in a greater temperature response (both globally and in the Arctic) than equivalent positive heat flux anomalies, as can be seen in the single-basin positive/negative heat flux simulations (Figs. 3 and 4; Table 1).

**Single-basin heat flux perturbations.** To elucidate the mechanisms by which the North Pacific dominates the Arctic temperature response, we evaluate the climate feedbacks involved in the single-basin heat flux experiments. The additional ocean-to-atmosphere heat flux of +1 W/m² global-equivalent from the North Pacific (NP-Warm) results in annual surface temperature increases ranging from 0.5 to 6°C across the entire northern hemisphere, with warming concentrated across the North Pacific Ocean and Arctic (Fig. 3). Annual-mean Arctic sea ice is reduced by 7.36 ± 0.54% relative to the control simulation, with the greatest decline focused along the sea-ice edge in the Greenland and Barents Seas and in the Sea of Okhotsk in the western Pacific (Fig. 3). Surface warming is most pronounced in winter months (December–February) and is amplified along the North Atlantic sea-ice edge and northern Siberia (Supplementary Figure 2), where surface temperature anomalies reach 6–7°C in winter. The mean global and Arctic temperature anomalies in the NP-Warm simulation are +0.95 ± 0.03 and +3.08 ± 0.20°C, respectively.

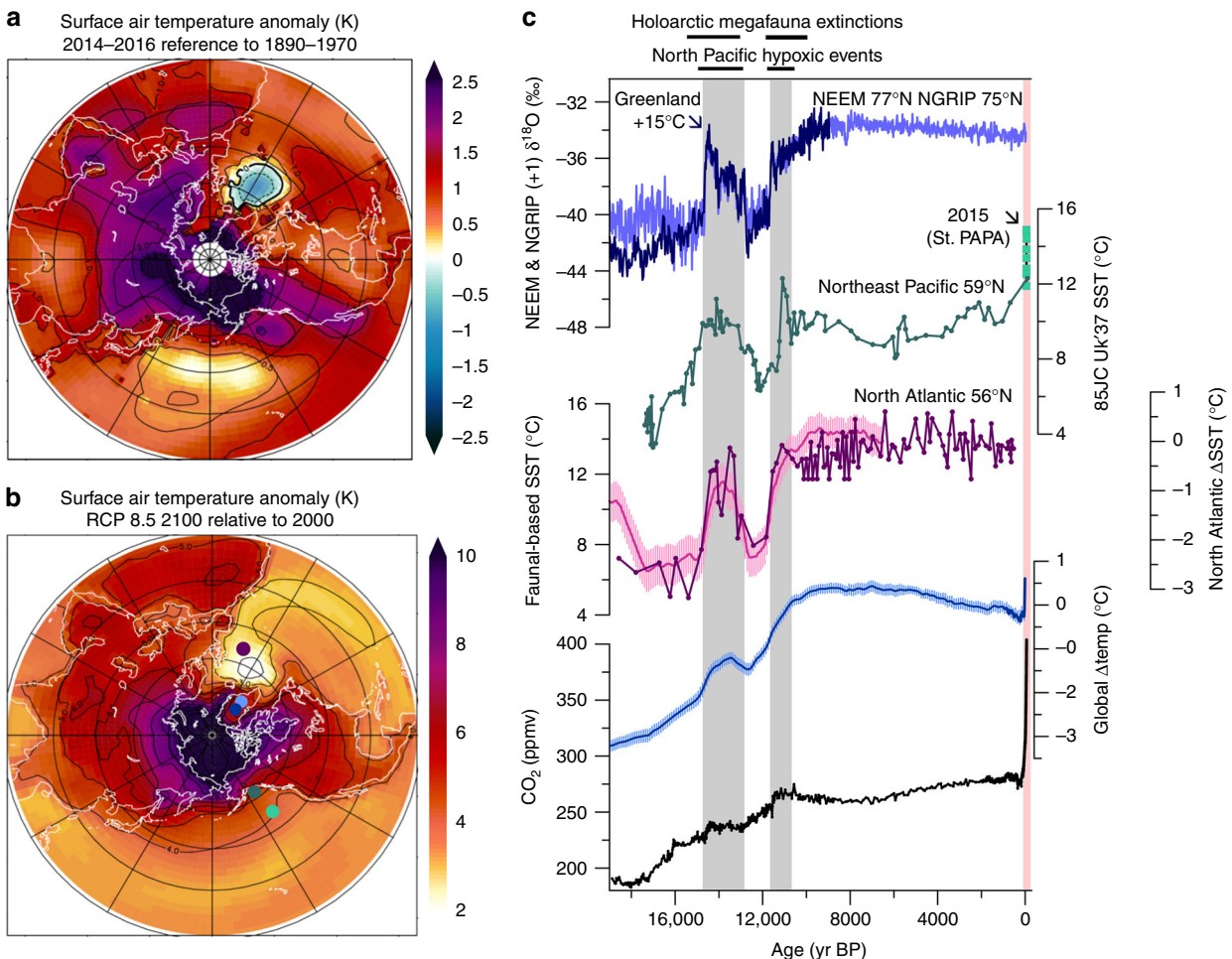

**Fig. 1** Recent and projected surface air temperatures in context of past abrupt warming events in the northern hemisphere. Recent annual surface air temperature (SAT) anomalies for 2014–2016 relative to 1890–1970 (**a**; refs.[58,59]), CMIP5 projected SAT anomalies by the end of the century in RCP 8.5 (**b**), and paleoclimate records spanning the past 19,000 years that document abrupt Arctic warming in concert with abrupt warming in the mid-latitude North Pacific and North Atlantic basins (**c**). Colored dots on map in **b** represent the locations for the paleoclimate datasets plotted in **c**. Record-breaking Arctic warming in recent years was associated with anomalously high SSTs in the Northeast Pacific despite a cold spot in the North Atlantic (**a**). Projected SATs show an amplified response in the Arctic and North Pacific, whereas North Atlantic SSTs undergo relatively modest warming in the first century (**b**). **c** Paleoclimate archives spanning the last 19,050 years (1950 = year 0). From top: Greenland δ[18]O records from NEEM 19,000–10,000 years ago (dark blue[26]) and NGRIP (light blue[60]), An alkenone-based UK′37 SST reconstruction from the Gulf of Alaska reflecting Summer SST (dark green[22]); the youngest data point represents the core top measurement taken in 2004 (ref.[61]); September SST data from Ocean Station Papa (145°W, 50°N) from 2007 to 2015 are plotted on the same axis (light green squares; http://www.pmel.noaa.gov/ocs/data). An SST reconstruction based on faunal assemblages from the central North Atlantic (plum[24]), and a reconstruction of the average North Atlantic SST anomaly relative to Holocene values (pink[25]). A composite record of the global temperature anomaly for the past 20,000 years relative to the 1980–2004 mean (blue[25,62,63]). Atmospheric $CO_2$ concentrations from 19,000 years ago to present based on ice cores measurement and modern measurements[64]. Gray shaded bars represent two episodes of rapid warming in Greenland that are coeval (within age uncertainties) with abrupt SST increase in the Northeast Pacific and North Atlantic, increases in $CO_2$ concentration, and intervals of widespread hypoxia in the North Pacific[22,28] and Holarctic megafauna extinctions[29]. Pink bar denotes changes since 1950

Overall, the NA-Warm simulation elicits a more muted northern hemisphere warming compared to the NP-Warm simulation, with reduced surface warming in the Arctic, Northeastern Pacific, and North American continent. Annual-mean SSTs increase by 5–6°C near the region of modified heat flux, with smaller magnitudes of warming seen elsewhere in the northern hemisphere (0.5–3°C) (Fig. 3). Annual-mean sea-ice area decreases by 4.98 ± 0.56% relative to the control simulation, with the greatest decline in the Beaufort Sea, along the perimeter of northern Alaska. The mean global and Arctic temperature anomalies in the NA-Warm simulation are +0.79 ± 0.02 and +2.12 ± 0.20°C, respectively.

**Sea-ice and low-cloud response.** In both the NP-Warm and NA-Warm simulations, Arctic sea-ice decline is most pronounced in regions with the greatest increase in low-cloud formation. Low clouds can be both a consequence and cause of sea-ice decline. Low clouds exert two opposing effects on the surface radiation budget; firstly, the enhanced albedo of cloud droplets decreases surface absorption of incoming shortwave radiation, and secondly, low clouds reduce outgoing longwave radiation[3]. Low-cloud formation in the subpolar extratropics results in a net surface cooling through the first mechanism. In the Arctic, however, enhanced low-cloud formation results in an overall surface warming effect in winter, spring, and autumn due to the

**Table 1 Summary of simulations conducted in this study**

| Simulation name | Q-flux modification (global-equivalent W/m²) | Topography modification | Global surface temperature (K) | Δ Global surface temperature (K) | Δ Arctic surface temperature (K) | Δ Arctic sea-ice fraction (%) | Δ Arctic low-cloud fraction (%) | Δ Arctic moisture convergence (%) | Arctic amplification factor | Climate feedback parameter α (W/m²K) |
|---|---|---|---|---|---|---|---|---|---|---|
| Preindustrial Control | — | — | 286.56 ± 0.02 | — | — | — | — | — | — | — |
| NP-Warm | North Pacific +1 | — | 287.51 ± 0.02 | 0.95 ± 0.03 | 3.08 ± 0.20 | −7.36 ± 0.54 | 12.4 ± 0.89 | 18.23 ± 2.47 | 3.25 | 1.05 |
| NEP-Warm | Northeast Pacific +1 | — | 287.47 ± 0.05 | 0.91 ± 0.24 | 2.55 ± 0.68 | −5.81 ± 1.51 | 9.99 ± 2.70 | 17.1 ± 6.25 | 2.80 | 1.10 |
| NA-Warm | North Atlantic +1 | — | 287.35 ± 0.02 | 0.79 ± 0.02 | 2.12 ± 0.20 | −4.98 ± 0.56 | 9.5 ± 0.87 | 15.14 ± 2.53 | 2.68 | 1.27 |
| NP-Warm/NA-Cool | North Pacific +1 North Atlantic −1 | — | 286.53 ± 0.02 | −0.030 ± 0.03 | 0.34 ± 0.22 | −0.31 ± 0.56 | −0.33 ± 0.93 | −0.05 ± 2.26 | — | — |
| NP-Cool/NA-Warm | North Pacific −1 North Atlantic +1 | — | 286.18 ± 0.03 | −0.39 ± 0.03 | −1.88 ± 0.14 | 4.07 ± 0.21 | 4.80 ± 0.29 | −4.84 ± 2.28 | — | — |
| NP-Cool | North Pacific −1 | — | 284.93 ± 0.02 | −1.63 ± 0.02 | −5.56 ± 0.15 | 9.96 ± 0.21 | −17.35 ± 0.29 | −24.69 ± 1.57 | 3.41 | 0.61 |
| NA-Cool | North Atlantic −1 | — | 285.45 ± 0.02 | −1.11 ± 0.02 | −3.29 ± 0.14 | 6.18 ± 0.22 | −11.65 ± 0.29 | −18.03 ± 1.62 | 2.96 | 0.90 |
| Control Notopo | — | Global land topography = 0 | 286.94 | — | — | — | — | — | — | — |
| NP-Warm-Notopo | North Pacific +1 | Global land topography = 0 | 287.97 | 1.03 | 3.23 | −7.33 | — | 18.60 | 3.14 | 0.97 |
| NA-Warm-Notopo | North Atlantic +1 | Global land topography = 0 | 287.81 | 0.87 | 2.43 | −4.90 | — | 16.36 | 2.79 | 1.15 |

Reported errors are the 95% confidence interval calculated as the standard error multiplied by 1.96, accounting for autocorrelation in time series (see Methods)

Surface air temperature anomaly (K)

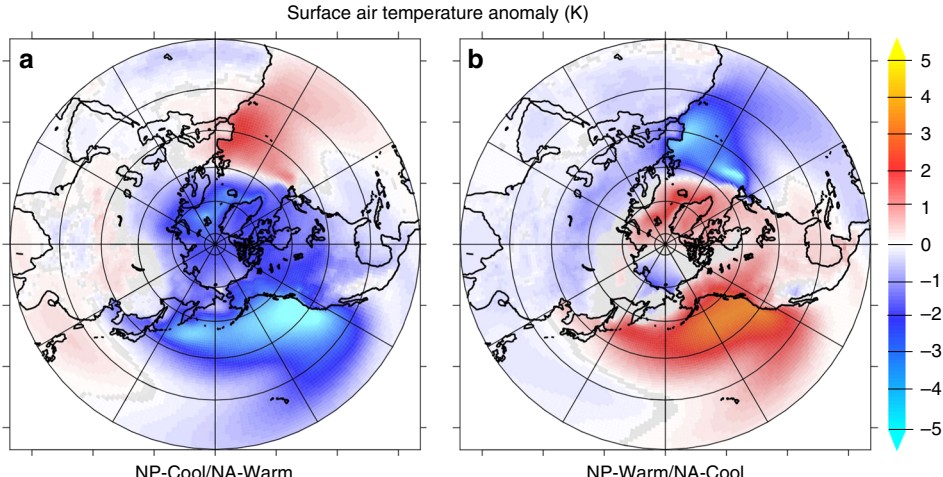

**Fig. 2** Surface air temperature anomalies in the zonally redistributed heat flux simulations. The NP-Cool/NA-Warm simulation (**a**) prescribes a negative heat flux perturbation in North Pacific (−1 W/m² global) compensated with an equivalent positive North Atlantic Q-flux perturbation (+1 W/m² global), such that the net global energy perturbation is zero. The sign of the heat flux perturbations in the NP-Warm/NA-Cool simulation (**b**) are reversed relative to the NP-Cool/NA-Warm simulation. In both simulations, the North Pacific heat flux anomaly dominates the surface air temperature response in the Artic. Gray areas indicate changes not significant at the 95% confidence level

second mechanism[32]. The net change in the cloud radiative effect at the top of the atmosphere may still be negative at high latitudes as a result of reflected incoming shortwave feedback and cloud brightening in response to increasing temperature[33], however, we focus here on the net radiative effects at the surface, given the influences on Arctic surface temperature and sea-ice extent.

The seasonal trends in sea-ice reveal a strong correlation between sea-ice decline and low-cloud formation, as the two patterns covary in all seasons except summer, when the radiative effect of low clouds reverses sign (Supplementary Figure 2, 3). The retreat of sea-ice has been shown to promote low-cloud formation through latent heat release and evaporation at the surface[34,35]; thus these two processes are likely coupled and reinforce one another. A linear regression of the change in sea-ice area and vertically integrated low-cloud fraction shows a high negative correlation between these variables in our simulations (Supplementary Figure 4).

In all simulations, increased ocean heat fluxes to the atmosphere drive decreases in mid-latitude cloud cover; changes of

opposite sign in heat flux have effects of the opposite sign in cloud cover (Figs. 3 and 4). In both the NP-Warm and NA-Warm simulation, the reduction in low clouds over the extratropical oceans increases shortwave absorption, further warming the sea surface and overlying air, decreasing relative humidity and increasing atmospheric moisture content that can be transported poleward.

Within the Arctic, the NP-Warm simulation increases the low-cloud formation more (+12.4 ± 0.89%) than the NA-Warm simulation (+9.5 ± 0.87%). The effect of low clouds accounts for an additional 2.9 ± 0.22 W/m² of surface longwave radiative forcing within the Arctic for the NP-Warm simulation versus an additional 2.2 ± 0.22 W/m² in the NA-warm simulation (calculated as the difference in downwelling longwave radiation at the surface between all-sky and clear-sky conditions). Clear-sky downwelling longwave radiation also increases more strongly in the NP-Warm case (10.7 ± 0.71 W/m²) than in the NA-Warm case (7.3 ± 0.69 W/m²). The enhanced longwave radiation contributes to surface warming and sea-ice retreat, increasing the net

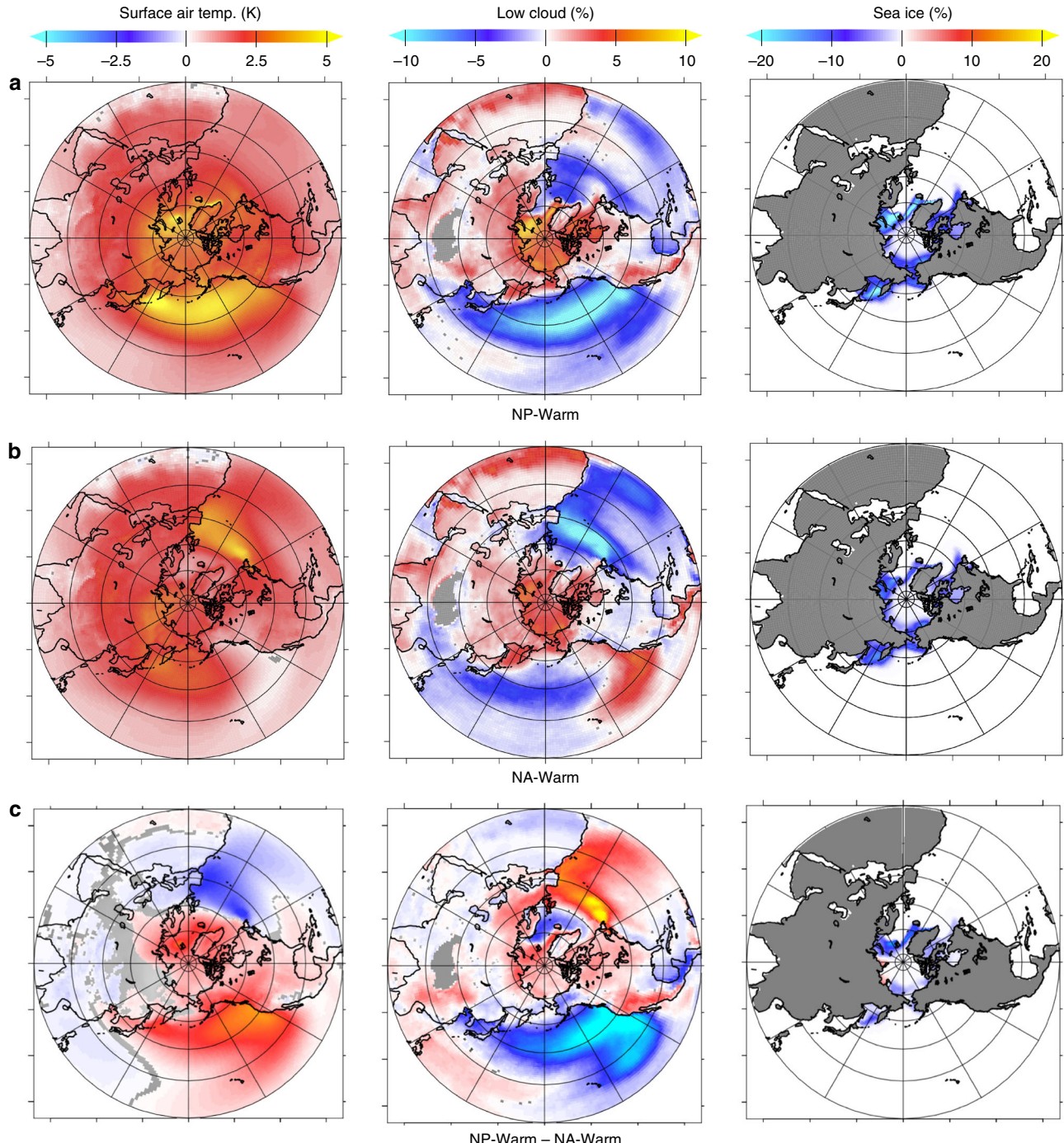

**Fig. 3** Surface air temperature, vertically integrated low cloud, and sea-ice anomalies in the warm ocean simulations. Simulations for **a** warm North Pacific (NP-Warm), **b** simulations for a warm North Atlantic (NA-Warm), and **c** the difference of the two simulations (NP-Warm − NA-Warm). Gray areas indicate changes not significant at the 95% confidence level

shortwave radiation absorbed at the sea surface through the reduction in surface albedo ($+1.7 \pm 0.42$ and $0.9 \pm 0.37$ W/m² for the NP-Warm and NA-Warm simulations, respectively).

**Atmospheric energy transport**. Decompositions of the total northward atmospheric energy transport into the dry static energy (sensible heat and geopotential energy terms) and moist (latent heat term) energy terms indicate that the total atmospheric northward energy transport into the Arctic is larger in the NP-Warm case (0.054 PW) than in the NA-Warm Case (0.043

PW), and that this difference is primarily driven by a larger increase in northward latent energy transport in the NP-Warm case (0.081 PW) than in the NA-Warm case (0.068 PW) (Fig. 5a). The increase in northward latent energy transport is partially offset by a decrease in dry static energy transport that is similar between the NP-Warm (−0.028 PW) and NA-Warm (−0.025 PW) cases. This translates to a total of 2.6 W/m² of direct Arctic heating from atmospheric energy convergence in the NP-Warm case and 2.1 W/m² in the NA-Warm case. Analysis of the longitudinal distribution of the northward latent energy transport

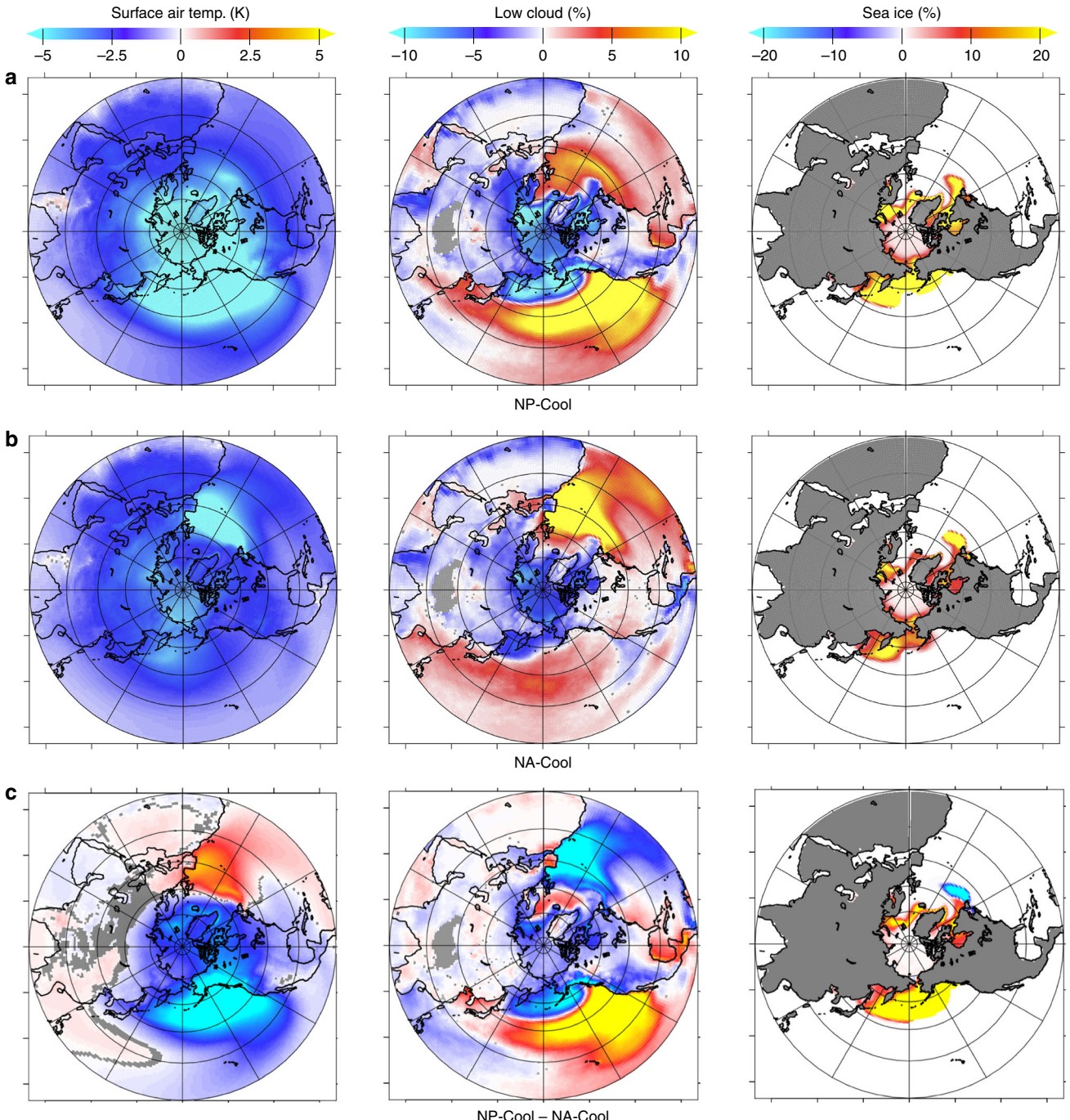

**Fig. 4** Surface air temperature, vertically integrated low cloud, and sea-ice anomalies for the cool ocean simulations. Simulations for **a** cool North Pacific (NP-Cool), **b** cool North Atlantic (NA-Cool), and **c** the difference of the two simulations (NP-Cool – NA-Cool). Gray areas indicate changes not significant at the 95% confidence level

along the Arctic boundary indicates the dominant regions that this latent energy is entering the Arctic in each case (Fig. 5b). In both the NA-Warm and NP-Warm cases, the total latent heat transport (i.e., amount of moisture intrusion) is enhanced across the Pacific and Atlantic basins relative to the control. The NP-Warm simulation shows a greater total latent heat transfer across the Arctic boundary through the Atlantic gateway, whereas the NA-Warm simulation shows greater total latent heat transfer over Siberia, relative to the NP-Warm case.

A further decomposition of the total atmosphere-integrated latent energy transport into the mean and eddy components show that the longitudinal structure of the change in total transport follows the change in mean transport (Supplementary Figure 5). The eddy transport term shows different longitudinal patterns for each case, with the NP-Warm case resulting in greater eddy transport across the North Pacific and North American continent, whereas the NA-Warm case shows an increase in eddy transport across most latitudes except the Northwestern

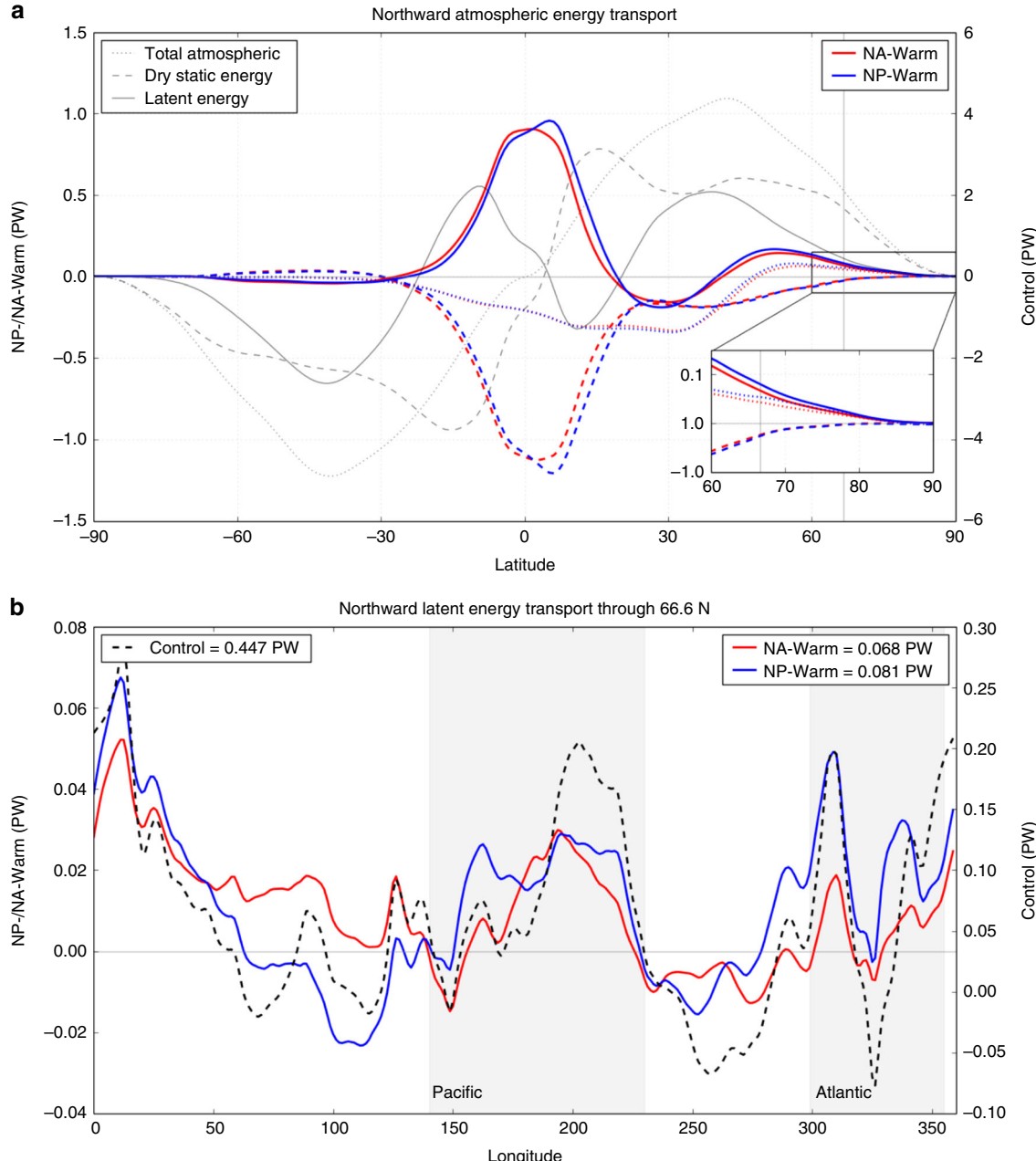

**Fig. 5** Decompositions of atmospheric energy transport. The total atmospheric energy transport across latitude, decomposed into dry static energy (sensible heat and geopotential energy terms) and moist (latent heat term) energy (**a**). Control values are in gray (as absolute values), NA-Warm anomalies relative to the control are in red, and NP-Warm anomalies are in blue. Inset shows a blow-up of the difference in the two cases from 60 to 90°N. The increase in latent heat transport dominates the total change in northward atmospheric energy transport, and is greater in the NP-Warm case relative to the NA-Warm case. **b** A zonal decomposition of the northward latent heat transport across the Arctic boundary (66.6°N), depicting the longitudinal distribution of the change in latent heat transport (i.e., moisture intrusions), with the Pacific and Atlantic basins shaded in light gray for reference. The values in the upper right corner depict the total latent heat transfer (integrated throughout the atmosphere) across the Arctic boundary. Control values are the dashed line (as absolute values), whereas the NA-Warm anomalies are in red, and NP-Warm anomalies are in blue

Pacific and eastern North American continent. The longitudinal integral of each term indicates that the mean and eddy changes contribute 22.9 and 77.1% of the total latent energy transport increase in the NA-Warm Case and 23.6 and 76.4% in the NP-Warm case.

## Discussion
The larger net moisture transport into the Arctic in the NP-Warm case results in enhanced Arctic surface warming through a

number of processes and feedbacks: greater latent heat transfer from the subpolar extratropics to Arctic, greater increase in the area of low clouds in the Arctic and attendant absorption of infrared radiation, and greater amounts of atmospheric water vapor in the Arctic and attendant clear-sky absorption of infrared radiation. The relative magnitude of these effects discussed above (2.6 W/m$^2$ of increased total atmospheric energy convergence, 2.9 W/m$^2$ of increased cloud-sky surface longwave radiation, 10.7 W/m$^2$ of increased clear-sky surface longwave radiation,

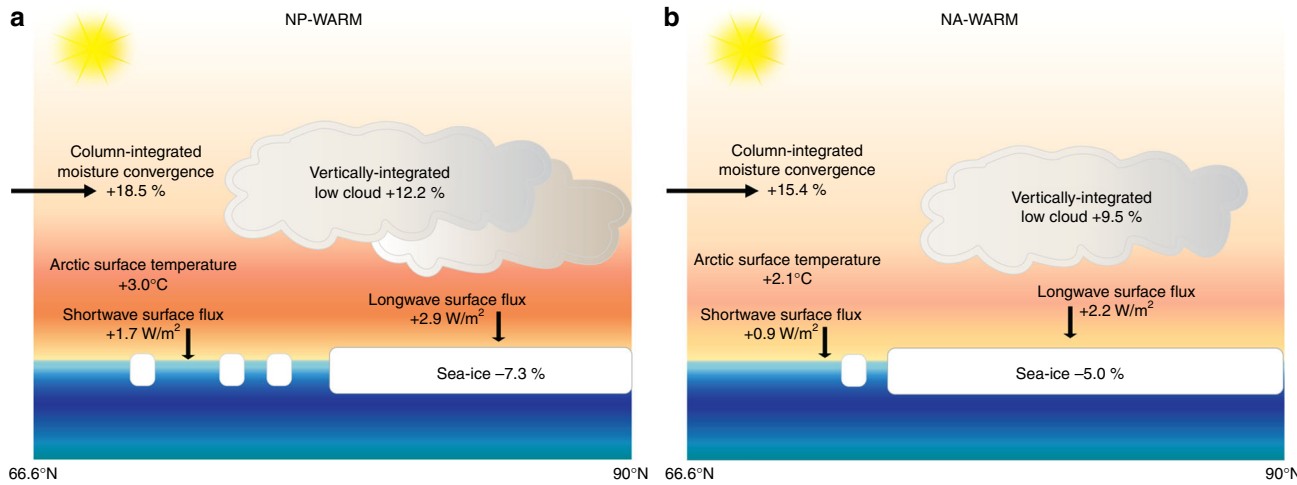

**Fig. 6** Schematic of major processes contributing to Arctic surface warming in the NP-Warm and NA-Warm simulations. Enhanced ocean heat flux increases the water vapor and latent heat transport into the Arctic, stimulating sea-ice retreat and an increase in low-cloud formation. The reported values are the mean anomalies integrated for the Arctic region for the NP-Warm simulation (**a**) and NA-Warm simulation (**b**)

respectively) suggests that all three mechanisms play an important role in Arctic warming. There is additionally greater sea-ice decline, which increases surface radiation absorption through reduction in surface albedo (ice-albedo feedback) (Fig. 6). As sea-ice retreats, evaporation increases, driving additional low-cloud formation and atmospheric moistening. Changes in moisture transport from the sub-Arctic to the Arctic thus plausibly provide the initial driver for Arctic change and these changes are then amplified by local processes.

What facilitates greater moisture transport into the Arctic in the positive North Pacific heat flux perturbation relative to the positive North Atlantic heat flux perturbation? Plausible mechanisms include: topographic mediation of atmospheric flow trajectories, the different areal extent of the North Atlantic and North Pacific basins and attendant impacts on surface albedo and air–sea moisture exchange, or the land–sea configuration and associated atmospheric pressure anomalies related to land–sea thermal gradients.

In order to test the first hypothesis, we ran identical simulations to the NP-Warm and NA-Warm cases with global land topography equivalent to a geopotential height of sea level (NP-Warm-Notopo, NA-Warm-Notopo). These simulations show an overall similar pattern to the heat flux perturbations with realistic land topography (Supplementary Figure 6), with surface air temperatures in the Arctic more strongly influenced by the positive heat flux anomalies sourced from the North Pacific (+3.23 °C) than from the North Atlantic (+2.43 °C) (Table 1). The reduction in topographic relief facilitates greater moisture transport from mid-latitudes to the Arctic, resulting in a slight increase in annual Arctic surface warming in both the NP-Warm-Notopo and NA-Warm-Notopo cases relative to the NP-Warm and NA-Warm cases, further supporting the central role of moisture transport as a major influence on the Arctic climate response. These results indicate that the atmospheric flow patterns influenced by land topography are not the causal mechanism for the greater moisture fluxes into the Arctic arising from the North Pacific heat flux anomalies (relative to the North Atlantic heat flux anomalies).

In order to test the second hypothesis, we reconfigured the North Pacific heat flux anomaly to an area equivalent to the North Atlantic heat flux anomaly (Supplementary Figure 7) and ran an additional simulation with a positive heat flux anomaly in the Northeast Pacific (NEP-Warm). The NEP-Warm simulation

results in a slightly reduced mean Arctic and global surface warming (2.55 ± 0.68 and 0.91 ± 0.24 °C, respectively) relative to the original NP-Warm simulation (3.08 ± 0.20 and 0.95 ± 0.03 °C, respectively), however, these changes are still greater than the equivalent area NA-Warm case (Table 1, Supplementary Figure 8). The larger errors associated with the NEP-Warm simulation in part reflects a more limited integration window (40–60 yr compared to 40–100 yr for the other simulations). However, the global temperature evolution of the NP-Warm and NEP-Warm simulations are statistically indistinguishable from one another throughout the entire overlapping simulation interval (1–60 yr: Supplementary Figure 9). This indicates that the greater area of the North Pacific heat flux relative to the North Atlantic heat flux anomaly in the original simulations does not have an appreciable effect on the resulting global and Arctic surface temperature response, at least in so far as we can ascertain from our simulations here.

Commonalities between the sea-level pressure anomalies in the simulations with no topography and realistic land topography provide insight into atmospheric pressure patterns that are primarily controlled by land–sea thermal contrasts rather than topography (the third hypothesis). Sea-level pressure changes in the NA-Warm simulation produce a high-pressure anomaly over the North Pacific that is most pronounced in winter (Supplementary Figure 10), indicating a general weakening of the Aleutian Low in response to a positive North Atlantic heat flux. A stronger Aleutian Low is associated with the transport of warm moist air from the North Pacific into the Arctic through the Bering Strait[36], whereas a weakened Aleutian Low shifts the North American ridge into the central Pacific, where it effectively blocks west-to-east propagating storm systems that carry moisture northward[37]. In contrast, the NP-Warm simulation produces a low-pressure anomaly that extends from the Northeastern Pacific into the Arctic during winter.

The NP-Warm-Notopo and NA-Warm-Notopo sea-level pressure anomalies result in a more homogenous northern hemisphere pattern than the NP-Warm and NA-Warm sea-level pressure anomalies (Supplementary Figure 11), which exhibit strong differences in their response over the North Pacific. Nevertheless, both the NP-Warm and NP-Warm-Notopo simulations do share a common low-pressure 'breach' along the Northeastern Pacific and North American continent that punctures the high-pressure ring encircling the Arctic from 45 to

60°N. In contrast, NA-Warm-Notopo exhibits a sea-level pressure pattern more in line with a negative phase of the Arctic Oscillation, with a continuous high-pressure ridge that helps to isolate the Arctic from subtropical heat and moisture intrusions.

We thus infer that the land–sea configuration is likely the primary mechanism that leads to greater moisture flux into the Arctic in response to the North Pacific heat flux anomalies. Some of the asymmetry in the Arctic climate response to North Pacific or North Atlantic heat fluxes in our simulations may in part be related to model biases or poor representations of cloud physics and shallow convective processes, such as the overly strong Pacific center of the wintertime Arctic Oscillation observed in many climate models[38], or the tendency for models to underestimate moisture intrusions to the Arctic through the North Atlantic gateway and overestimate moisture intrusions from the Pacific sector[39]. However, the fact that much of the total latent heat energy entering the Arctic is channeled through the North Atlantic gateway in response to North Pacific warming (NP-Warm; Fig. 5b) makes it unlikely that this latter factor[39] can fully account for the asymmetries in the poleward moisture transport in the NP-Warm and NA-Warm simulations.

Due to the stronger Arctic temperature response elicited from the North Pacific (in comparison to North Atlantic) heat flux anomalies, the global temperature response is more sensitive to North Pacific warming and cooling, implying enhanced climate sensitivity to SST perturbations in this region (Table 1, Supplementary Figure 9). However, our simulations do not have bearing on the sensitivity of the Arctic to changes in heat fluxes northward of 60°N, which is today a region of high ocean heat flux due to deep water formation in the Nordic Seas, and may thus be a potentially dynamic region for changes in ocean heat flux in the past[40]. Nevertheless, changes in ocean heat flux sourced from the North Pacific have been largely overlooked as a mechanism for Arctic change in the past, in part due to a weaker modern overturning circulation compared to the Atlantic, which results in smaller magnitudes of ocean-to-atmosphere heat flux (Supplementary Figure 1). However, both paleoclimate data and modeling studies suggest the patterns of ocean heat flux and circulation between the North Atlantic and North Pacific may have been significantly different or even reversed in the past[41]. Our simulations suggest that more diffusely distributed ocean heat fluxes in the North Pacific may have the capacity to exert stronger influences on global and Arctic climate than more concentrated heat fluxes in the North Atlantic through modulations in low-cloud cover and poleward moisture transport.

While the slab-ocean configuration limits the assessment of ocean feedbacks that may amplify or dampen the atmospheric feedbacks outlined here, these simulations highlight the strong downwind impacts of ocean thermal anomalies; SST changes in the North Pacific can have substantial impacts on the Atlantic sector of the Arctic, and SST changes in the North Atlantic can influence the Northwestern Pacific and Pacific sector of the Arctic. Our results are broadly consistent with surface air temperature anomaly patterns generated in a coupled general circulation model in response to forced SST cooling in the North Pacific[42]. We note that North Atlantic heat flux anomalies result in relatively weak SST anomalies in the Northeastern sector of the Pacific in our study, consistent with many model studies that exhibit either weak or even reversed temperature anomalies in the Northeastern Pacific in response to forced perturbations in North Atlantic circulation[41,43]. In contrast, past SST reconstructions suggest that SST changes of similar magnitudes between the North Atlantic and Northeastern Pacific occurred in conjunction with abrupt Arctic changes[21–23] (Fig. 1c), challenging the pervasive assumption that abrupt North Pacific warming events in

the past were merely a downwind response to North Atlantic ocean circulation changes. Collectively, these data and model results may instead point towards unidentified mechanisms of North Pacific "regime shifts" that result in amplified SST warming along the Northeastern Pacific.

Models predict the greatest SST increases in the North Pacific in the coming century, with SSTs up to 3°C greater than equivalent latitudes in the North Atlantic, attributed to the shallow North Pacific mixed layer depth, which allows a more rapid equilibration with atmospheric temperature increase[19]. This "North Pacific Rapid Response" may help to provide a mechanistic framework for understanding the initiation of rapid SST changes in the North Pacific that coincide with the timing of rapid pulses of atmospheric $CO_2$ in the past, as well as the close coupling of SSTs in the Northeastern Pacific with trends in atmospheric $CO_2$ over the past 18,000 years[22] (Fig. 1c). The North Pacific Rapid Response may act to propagate and amplify the signal of radiative forcing through close coupling with Arctic amplification feedbacks.

Our simulations suggest that surface ocean warming in the North Pacific may have a particularly pronounced effect on Arctic temperature and sea-ice extent, and may help to explain the record-breaking Arctic surface warming and sea-ice decline in recent years[16] that accompanied unusually warm ocean temperatures in the Northeast Pacific[17]. Observational records indicate low clouds have been increasing in the Arctic in the past few decades, especially in the Beaufort-Laptev region, and are associated with declines in sea ice and warmer surface air temperatures in winter, spring, and autumn[34,35]. The North Pacific has been identified as the dominant moisture source to this region of the Arctic in recent decades[44], where enhanced moisture transport is associated with accelerated sea-ice retreat[45,46]. Furthermore, the unprecedented decline in Arctic sea-ice extent in 2007 has been linked to anomalous heat and moisture intrusions from the Northwestern Pacific, through an accompanying increase in downwelling longwave radiation[47].

In summary, our experiments reveal a strong global and Arctic climatic dependence on the magnitude of water vapor and latent heat transfer from the mid-latitude oceans to the Arctic, with accompanying cloud, ice-albedo, and temperature responses that enhance Arctic warming. This linkage implies that projections of Arctic climate are highly sensitive to subpolar extratropical surface ocean temperature changes, making it essential to accurately estimate the magnitude of SST increase likely to occur in northern hemisphere oceans in response to rising greenhouse gases. Systematic cold biases in North Pacific and North Atlantic SSTs in CMIP5 models[48] may thus partly lead to an underestimation of Arctic warming and sea-ice decline in climate projections, with important ramifications for climate and ecological tipping points in the Arctic.

## Methods

**Model simulations**. We employ simulations with the National Center for Atmospheric Research's Community Earth System Model (CESM) version 1.0.4[30], which incorporates the Los Alamos Sea Ice Model version 4 (CICE4[49]), the Community Land Model version 4 (CLM4[50]), and the Community Atmosphere Model version 4 (CAM4[51]), coupled to a slab ocean. The atmosphere and land components are simulated on a 0.9° × 1.5° grid, whereas the ice and ocean components are on an approximately 1° × 1° grid. A total of 12 simulations are conducted under preindustrial atmospheric $CO_2$ concentrations (284.7 ppm) with various modifications to the ocean surface or mixed layer heat flux (Q-flux) and land topography (Table 1).

We use the routine available in the standard CESM download, which uses monthly oceanic fields of temperature, salinity, surface velocities, boundary layer depth, surface heat flux, melting and freezing fluxes to generate the Q-flux for a specific mixed layer depth. The mixed layer is an annual average to balance the seasonal cycle[52]. We use an 80-year average from a multi-millennia coupled control simulation with preindustrial forcing to generate a climatological reference state, which globally and annually average to a zero Q-flux. The annual net Q-flux is a

small residual of a strong seasonal cycle, with the same pattern as the surface heat flux, only with a smaller amplitude. In the equilibrium state, the heat flux at the surface and mixed layer depth are the same, since the slab-ocean simulation does not allow lateral transport. The annual-mean climatological Q-flux compares well with observations and is dominated by tropical upwelling regions taking up heat and Western boundary currents, the Nordic seas, and the Southern Ocean releasing heat to the atmosphere[31,53].

We then apply the globally non-zero Q-flux pattern as a negative (i.e., heat flux out of the mixed layer depth into the ocean) and positive (i.e., heat flux into atmosphere) ocean heat uptake forcing inducing a global warming and cooling. The idealized Q-flux anomaly patterns are sine-shaped functions being bound to zero at 30°N and 60°N and rising to a maximum value at 45°N. The global imbalance was tuned to +1 W/m² and −1 W/m² global-equivalent, which implies the Pacific local maximum is smaller (31.5 W/m²) than the Atlantic local maximum (54 W/m²), since the Pacific area is larger (Supplementary Figure 1). We also configured a Q-flux anomaly for the North Pacific that was the equivalent area as the heat flux imposed on the North Atlantic to eliminate the potential effects of the different areas of the North Pacific and North Atlantic heat flux anomalies on global and Arctic climate. The local anomalous Q-flux is in the same order of magnitude as the climatological annual-mean flux, but smaller than the local seasonal cycle. The mixed layer depth is kept at the climatological control simulation annual average (spatially varying).

The naming convention for the simulations is as follows: NP-Warm (North Pacific +1 W/m²), NP-Cool (North Pacific −1 W/m²), NA-Warm (North Atlantic +1 W/m²), NA-Cool (North Atlantic −1 W/m²) for the single-basin Q-flux perturbations, and NP-Warm/NA-Cool and NP-Cool/NA-Warm denoting the compensating "seesaw" Q-flux perturbations. Simulations were also run with the positive ocean-to-atmosphere Q-flux perturbations in each basin with modified land topography, such that the global land surface was set to a geopotential height of zero (NP-Warm-Notopo and NA-Warm-Notopo). A simulation was run with a positive heat flux imposed in the Northeast Pacific that was approximately equivalent to the area of the Atlantic heat flux (NEP-Warm).

**Model output analysis**. All simulations were run for 100 model years and annual and seasonal averages were calculated for years 40–100 of the model output, with the exception of the NEP-Warm simulation, which was run for 60 model years with mean values calculated from years 40 to 60. Reported errors are calculated as the standard error of the mean multiplied by 1.96, which is the number of standard deviations required to contain 95% of the area of a normal distribution. The calculation of the standard error of the mean assumes a sample size of $N$ independent samples, which in our case are the individual model years. Autocorrelation between model years, however, reduces the number of independent samples, as described by Santer et al.[54] This autocorrelation is corrected for by using Eq. (6) of Santer et al.[54] to determine an "effective sample size", $N_e$, that adjusts the original sample size for this autocorrelation.

We calculate the global climate feedback parameter ($\alpha$) as $F/\Delta T$, where $F$ is the applied global radiative forcing ($\pm 1$ W/m²) and $\Delta T$ is the change in global average surface air temperature[55]. The Arctic amplification factor is calculated as the ratio of the change in global surface air temperature to the change in Arctic surface air temperature ($\Delta T_{\text{global}}/\Delta T_{\text{arctic}}$) (Table 1). The cloud radiative forcing at the TOA is calculated as the difference in the net radiative flux between all-sky and clear-sky conditions. The greenhouse effect of clouds in the Arctic is calculated as the difference in the downwelling longwave radiation at the surface between all-sky and clear-sky conditions[56]. We calculate the surface albedo forcing related to changes in sea-ice as the difference in upwelling shortwave radiation at the surface in clear-sky conditions. However, the enhanced absorption of shortwave radiation at the surface due to sea-ice retreat is in part compensated by enhanced shortwave reflection at the TOA due to changes in Arctic cloudiness. For example, the greater increase in Arctic cloud fraction in the NP-Warm contributes to an annual-mean negative cloud radiative effect at the top of the atmosphere (TOA) (−2.0 ± 0.36 and −1.7 ± 0.26 W/m² for the NP-Warm and NA-Warm simulations, respectively). However, this is composed of a positive cloud radiative effect from trapped longwave in winter—when the maximum Arctic warming occurs—and a negative cloud radiative effect from reflected shortwave in the summer, during which season the decreased albedo from sea-ice loss more than compensates for the increased cloud reflection. Thus, the warming effect of low clouds and sea-ice retreat together more than compensate for the reduction in incoming shortwave at the TOA due to enhanced cloud albedo.

The net change in shortwave radiation at the surface is interpreted as the effective shortwave absorption due to both cloud and sea-ice changes. Moisture convergence in the Arctic is defined as precipitation minus the surface water flux. Changes in Arctic sea-ice and cloud fraction are reported as the percent change in the area fraction relative to the control simulation. All calculations for the Arctic are defined as north of 66.6°.

The total northward atmospheric energy transport was calculated at each latitude, as well as its decomposition into dry static energy (i.e., sensible heat and geopotential energy terms) and moist (i.e., latent heat term) energy, following Yang et al.[57] As in Yang et al.[57], the total atmospheric energy transport through each

latitude ($H_a(\Phi)$) is calculated as:

$$H_a(\phi) = 2\pi a^2 \int_{-\pi/2}^{\phi} \cos\phi'(R_{\text{TOA}} + F_s)\,d\phi'$$

for Earth's radius ($a$), latitude increment ($\Phi'$), top-of-atmosphere radiative flux ($R_{\text{TOA}}$), and surface radiative ($F_S$), with sign convention such that $R_{\text{TOA}} + F_S$ constitutes energy into the atmosphere. The latent heat transport through each latitude ($H_{\text{LH}}(\Phi)$) is similarly calculated using the moisture balance equation as

$$H_{\text{LH}}(\phi) = 2\pi a^2 \int_{-\pi/2}^{\phi} \cos\phi' L_v(\text{Evap} - \text{Precip})\,d\phi'$$

for latent heat of vaporization ($L_v$). The dry static energy transport is calculated as the residual, $H_A(\Phi) - H_{\text{LH}}(\Phi)$. Each term in the climatology as well as the perturbation in NP-Warm and NA-Warm cases is shown in Fig. 5a.

In order to diagnose where the latent energy is entering the Arctic in the NP-Warm and NA-Warm simulations, we analyzed the longitudinal distribution of latent energy transport across 66.6°N (Fig. 5b). We calculate this as the longitudinally resolved latent energy component of the vertically integrated moist static energy transport:

$$H_{\text{LE}} = \int_{p_t}^{p_s} aL_v \text{vq}\cos\varphi/g\,dp$$

integrating between atmospheric pressure at the surface ($p_s$) and top-of-model ($p_t$) at latitude ($\varphi = 66.6$ N), gravitational acceleration ($g$) and other variables as in previous equations. The moisture transport (vq) is calculated using the model online diagnostic VQ, which outputs the multiplication of the two terms at each model time step and includes both the mean and the eddy components of the latent energy transport (Supplementary Figure 5). The direct Arctic heating from atmospheric energy convergence is calculated as the total atmospheric northward energy transport values divided by the area of the Arctic ($2.089 \times 10^{13}$ m²).

**CMIP5 calculations**. The Coupled Model Intercomparison Project 5 (CMIP5) surface air temperature anomalies for 2080–2100 (Fig. 1b) in the RCP 8.5 scenario were calculated as an average of the following 23 models relative to their average surface air temperature of the historical simulations: ACCESS1-0, ACCESS1-3, bcc-csm1-1-m, bcc-csm1-1, BNU-ESM, CanESM2, CCSM4, CESM1-BGC, CESM1-CAM5, CNRM-CM5, CSIRO-Mk3-6-0, EC-EARTH, FIO-ESM, GFDL-CM3, GFDL-ESM2M, GFDL-ESM2G, GISS-E2-H, GISS-E2-R, IPSL-CM5A-LR, MIROC-ESM, MRI-CGCM3, NorESM1-M, NorESM1-ME.

The surface heat flux anomalies for RCP 8.5 were calculated using one model simulation from 37 CMIP5 models, based on the time-mean of years 2081–2100 in the RCP 8.5 scenario relative to the time-mean for the preindustrial control run over the first 200 years (Supplementary Figure 12).

Map figures were generated using the PyFerret software from NOAA's Pacific Marine Environmental Laboratory and the Panoply software (version 4.6.2) developed by the NASA Goddard Institute for Space Studies.

**Code availability**. The code and model output generated during the current study are available from the corresponding author upon reasonable request.

**Data availability**. The datasets generated during the current study are available from the corresponding author upon reasonable request.

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

## Acknowledgements

We thank Patrick Brown for assistance with Supplementary Figure 12, Anna Possner for helpful discussions, Jay Alder for comments that improved this manuscript, and the OCS Project Office of NOAA/PMEL for access to SST data from Ocean Station PAPA.

## Author contributions

S.P., M.R., and K.C. designed the experiment and ran the simulations. G.P. contributed to data processing and figure generation. All authors contributed to data interpretation. S.P. wrote the paper with contributions from G.P., M.R., and K.C.

## Additional information

**Competing interests:** The authors declare no competing interests.

