## [Peer Review File · Nature Communications]

Reviewers' comments:

Reviewer #1 (Remarks to the Author):

Review of "North Pacific surface ocean heat fluxes enhance Arctic warming" by S. Praetorius, M. Rugenstein, G. Persad, and K. Caldeira.

This manuscript starts from the observation that historical sea surface warming patterns are asymmetric between the N. Pacific and N. Atlantic, with strong warming in the subpolar Northeast Pacific. Various paleo records from the past 18,000 years show that both the N. Atlantic and N. Pacific SSTs increased roughly coevally with temperatures in Greenland during periods of abrupt warming. The authors conduct idealized experiments with a slab ocean configuration of the CESM climate model. The experiments are designed to measure differences in the climatic response to imposed sea surface heat fluxes in the extratropical regions of the two ocean basins. The major finding is that Arctic temperatures are more sensitive to imposed heat flux anomalies in the Pacific than the Atlantic.

The main result is very interesting and worth understanding in detail. Unfortunately I find some serious flaws in this manuscript and I can't recommend it for publication in its current form. My main criticism of the manuscript centers around Figure 4, and the conclusion that differences in moisture flux are the key driver of the differences in Arctic climate response to the Pacific and Atlantic perturbations (e.g. line 20-22).

The conclusion about the moisture flux seems to rely almost entirely on the finding that the zonally integrated moisture flux across 67°N increases more in the NP-WARM case than the NA-WARM case. There are a number of problems with this line of argument, which I outline here.

- First, the presentation is somewhat convoluted and I had a difficult time understanding what the authors were trying to say. For one thing, the numbers in the lower panel of Fig. 4 (+21%, +17%) do not agree with Fig. 4 or Table 1 (+18%, +15%).

- The discussion on lines 151-153 is confusing. Fig. 4 shows that there is enhanced moisture flux through the Pacific gateway in BOTH cases, and about the same magnitude.

- Line 155: "The relative troposphere-integrated Arctic air temperature changes ... are proportional to the zonal profiles of moisture flux. This supports a dominant role of water vapor and latent heat flux as the primary drivers of the Arctic temperature response"

I had a lot of trouble understanding what the authors meant here. First I assumed that it was a statement about the zonal structures of the temperature and moisture flux anomalies. But looking at the top and bottom panels of Fig. 4, I don't see any such proportionality.

I concluded that the authors were simply referring to the integrated values of Arctic surface temperature and moisture flux listed in Table 1. In this case, yes it is true that both the total moisture flux and Arctic warming are larger in the NP-Warm case.

In any case, this is not a strong argument for causality. All else equal, the moisture flux will increase with warming just due to the increase in saturation, regardless of the mechanism driving the warming.

- The authors compare sensible heat fluxes and moisture fluxes in a qualitative way. Unfortunately I think this analysis is incomplete and may be misleading. Fluxes of sensible heat and moisture are both important because they both contribute to the net transport of energy into the polar cap. The relevant metric for atmospheric heat transport is the flux of moist static energy (MSE), which includes the geopotential energy term in addition to sensible and latent heat terms:

$$\text{MSE} = c_p T + L q + g Z$$

where Z is the geopotential height. The fluxes in Fig. 4 only show the first two terms. On the other hand, analysis such as Overland and Turret (1994) show that the geopotential term is the largest

contribution to the MSE flux across 70°N in the lower troposphere.

But even without this information, we can quantitatively compare the relative contributions of sensible and latent heat to the anomalous heating of the Arctic cap by putting both fluxes into common energy units, simply by multiplying sensible heat by $c_p = 1004 \text{ J / kg / K}$ (specific heat at constant pressure), and multiplying latent heat by $L = 2.5 \times 10^6 \text{ J / kg}$ (latent heat of vaporization).

From Fig. 4, a typical magnitude of latent heat flux anomaly is about $0.6 \times 10^{-2} \text{ kg/kg m/s}$ --- yielding an anomalous energy flux of $1.5 \times 10^4 \text{ J/kg m/s}$

The zonal average sensible heat flux anomaly is harder to judge eye from Fig. 4, but might be about $0.5 \times 10^3 \text{ K m/s}$ --- yield an anomalous energy flux of $0.5 \times 10^6 \text{ J/kg m/s}$ --- two orders of magnitude larger than the moisture flux contribution! This doesn't seem plausible but is consistent with the values reported in Figure 4. And it casts serious doubt on the mechanism proposed by the authors. I strongly recommend that the authors do a more careful analysis of the different contributions to the poleward energy flux.

- A related point: I didn't find any information in the methods or supplemental material about how the fluxes were calculated. Are they based on online diagnostics where the correlations (e.g. v^*T , v^*q) are computed at every timestep in the model? Or are they computed offline by multiplying time-averaged v and T fields? Hopefully it is the former, but if the latter, it is critical to document the time frequency of the averaged output that was used to compute the correlations. Also, these are reported as "tropospheric averages". The authors need to be more specific somewhere in the methods about what vertical levels were included in the calculation.

Also, these kind of flux diagnostics are notoriously susceptible to spurious mass imbalance issues. The CESM uses a hybrid sigma-pressure vertical coordinate that introduces some complexity to the tropospheric diagnostics, particularly in the vicinity of topography. See appendices in Hill et al. (2015) and Yang et al. (2015) for discussion of the effects of spurious mass imbalance issues on energy flux diagnostics (and how to correct for them) in the GFDL and CESM models respectively.

- It's also possible that the energy flux diagnostics are a red herring here, and that the key role of increased moisture fluxes is actually to increase the cloud cover over the Arctic -- making the clouds the "local driver" of the warming rather than a feedback upon warming from enhanced energy transport. Testing this hypothesis would require a set of "locked cloud" simulations, which could shed very interesting light on the mechanisms.

- Regarding the mechanism for the different sensitivities to Pacific vs. Atlantic perturbations: the authors tested the hypothesis about topographic constraints, and found a most null result. I suggest considering a simpler hypothesis about the relative sizes of the Atlantic and Pacific basins. Because the Pacific is wider, the imposed q -flux (Fig. S1) is locally weaker in this case. In a perfectly linear climate system this shouldn't matter since their area-integrated values are identical. However there may be some non-linear effects operating at the sea surface such that the locally weaker Pacific perturbation results in a more "purely evaporative" response. With the clarity of hindsight, it might have been better to set up the experiments with identical q -fluxes in the two basins, with a locally larger Pacific q -flux covering an Atlantic-sized sector on the eastern side of the basin. This would have been a better-controlled experiment, and also quite relevant because of the role of Northeast Pacific SSTs in the observed anomalies in Fig. 1A.

Minor comments:

Line 12: shouldn't poleward atmospheric heat transport also be mentioned here?

Line 50: "ocean SST anomalies of +4°C in the North Pacific" ... This is inconsistent with Fig. 1a, which shows SAT anomalies that are mostly $< +2^\circ\text{C}$

Line 62: The text here should clarify that the "North Atlantic" in this case is a location nowhere near the "warming hole" that is so prominent in Fig. 1

Line 70: The text seems to draw an equivalency between Greenland temperatures and Arctic temperatures as a whole.

Line 81: It's not clear a priori that the important teleconnections linking the midlatitude sea surface to the Arctic will be manifest as changes in meridional heat transport.

Line 125: The authors are implicitly assuming that the cloud radiative effect is proportional to changes in cloud amount. However it is important to recognize that this is often not true for the SW effects of low clouds in cold regions. The reason is that the brightness of cold clouds increases strongly with temperature. This is a well-documented physical effect that is also represented in the models, and is a reason why SW cloud feedback tends to be negative at high latitudes (e.g. Zelinka and Hartmann 2012). I'm not sure whether this is an important issue in the present study, but should be acknowledged that cloud amount may be a poor proxy for SW cloud feedback in the Arctic.

Line 135: I appreciate how the authors are carefully avoiding assigning causality here. (This is a compliment, not a criticism)

Line 154: "troposphere-integrated Arctic air temperature" If I understood correctly, what's plotted in Fig. 4 are temperatures at 66°N. The wording seems to imply something like an area-integrated temperature over the Arctic region, which is not the same thing.

Line 180: "Changes in moisture transport from the sub-Arctic to the Arctic provide the initial driver for Arctic change..."

See major comments above. This is a reasonable hypothesis but I don't think it has actually been shown here.

Line 209: "our results challenge the pervasive assumption that abrupt North Pacific warming events in the past were merely a downwind response to North Atlantic ocean circulation changes" This seems like an overstatement. I don't see anything here to suggest that the hypothetical North Pacific driver would provide a better explanation or fit to the data.

Line 565-568: These arguments seem inconsistent with the meridional wind and moisture flux anomalies that are actually shown in Fig. 4.

References

J. E. Overland and P. Turet. Variability of the atmospheric energy flux across 70°N computed from the GFDL data set. In O. Johannessen, R. Muench, and J. Overland, editors, *The Polar Oceans and Their Role in Shaping the Global Environment*, number 85 in *Geophysical Monographs*, pages 313–325. American Geophysical Union, 1994.

S. A. Hill, Y. Ming, and I. M. Held. Mechanisms of forced tropical meridional energy flux change. *J. Climate*, 28:1725–1742, 2015.

H. Yang, Q. Li, K. Wang, Y. Sun, and D. Sun. Decomposing the meridional heat transport in the climate system. *Climate Dynamics*, 44:2751–2768, 2015.

M. D. Zelinka and D. L. Hartmann. Climate feedbacks and their implications for poleward energy flux changes in a warming climate. *J. Climate*, 25:608–624, 2012.

Reviewer #2 (Remarks to the Author):

Peer review of "North Pacific surface ocean heat fluxes enhance Arctic warming" by Praetorius et al.

The manuscript concerns the effect on the Arctic climate of temperature anomalies in the North Pacific. Based on model experiments it is argued that a heat flux anomaly in the North Pacific has a much larger impact on Arctic climate than has a similar anomaly in the North Atlantic region. This is a very interesting manuscript that provides a relevant and important contribution to the discussion on Arctic climate change attracting considerable attention these years. The study is comprehensive and creatively executed, and the methods appear robust. I have some minor points and suggestions, and recommend publication of these have been considered.

Points for the authors to consider:

1. The weakest point, in the evidences provided by the author for the importance of the North Pacific heat anomaly, is that the evidences mostly rely on a single climate model, although this model arguable is one of the worlds leading and most state-of-the-art models. I will still recommend to bring forward towards the end of the paper a caution of the results relying on the dynamics of a single climate model.

2. Since the fundamental idea is to investigate the effect of change of the ocean circulation, it seems more appropriate to construct all experiments with a zero forcing, as, fundamentally, the ocean circulation redistribute energy but does not produce energy. The sea-saw experiments are constructed that way, but those with forcing in a single ocean are not. It would be more relevant for a comparison across experiments that all experiments have zero forcing. Many earlier work have shown that Arctic amplification appears just as a result of a forcing, but here the intention seem to be to show how Arctic amplification is related to a given ocean heat-flux anomaly. However I am not requiring the authors to redo their experiments, only I am asking the authors to consider what the difference in forcing/no forcing implies for their results in this context.

3. L11-12: Perhaps mention atmospheric circulation change as an additional source of Arctic amplification (e.g. Graverson et al. Nature, 2008).

4. L24: "imply global" -> "imply that global".

5. L34: "Thought to be most" -> "often argued to be the most".

6. L96-99: Mention here that this sentence refers to experiments with forcing in a single ocean in contrast to the sea-saw experiments discussed just above. In order to see the difference in response to positive and negative forcing more clearly, as discussed in this sentence, difference plots could have been provided in Fig. 3. The sentence in L100-101 could be deleted since this point is likely clear to the reader at this point.

7. L168-169: The parenthesis can be omitted.

8. L169-172: Perhaps provide a reference to Fig. 4 after this sentence.

9. L178: How does latent heat advection reduces the Planck feedback? Please provide a sentence for explanation.

10. Fig. 1a and b: Continents are difficult to see. Perhaps provide the coast lines in white on top of the shading.

11. Fig. 4 caption: Please indicate what the numbers in the lower right corner of the frames

mean.

12. L461: "that" -> "a"?

13. L465: "which follows" -> "with the same pattern as".

14. L466-467: I don't understand this sentence, saying that slab-ocean does not allow lateral transport. As far as I know, the Q-fluxes are indeed the lateral transport.

15. L479: Perhaps remove the parenthesis.

16. L488-490: I Could not follow how this error was estimated. Please provide a sentence or two for clarification.

17. Fig. S1: Why is different types of map chosen (upper panel shows Pacific in the center, whereas the others show Africa)?

18. Fig. S7: Also here it would be appropriate to show difference plot for the context of the discussion in L541-546.

Reviewer #3 (Remarks to the Author):

Review of North Pacific surface ocean heat fluxes enhance Arctic warming by Praetorius et al. In this paper the authors argue that atmospheric energy transport changes are crucial in Arctic warming (which is well known), and that the Pacific plays a major part in this (which is also well known).

What is new, however (and should be reflected in the title), is that they demonstrate the Arctic (and global system) appears more sensitive for Pacific heat flux anomalies than for Atlantic ones, even if the global integrated change in ocean heat uptake/release anomaly remains the same. As a result they argue that the Arctic and global climate sensitivity is sensitive to the projected zonal distribution of heat flux anomalies.

Despite a host of caveats, which I will detail below, I think the results, even if the experimental set-up may appear unrealistic, are surprising and insightful, and therefore I recommend publication in Nature Comm after the authors have addressed the comments below.

- The experimental set-up was unclear to me. First, is the Q-flux as shown in Fig. S1 fixed? If so, the runs are essentially atmosphere only runs and the slab ocean is only used as a diagnostic to derive the Q-flux forcing? If not, please explain in more detail. Also, I assume the Q-fluxes were derived accounting for ocean heat transport divergence. If not the Q-fluxes applied are wrong (biased). Please explain this better and try to estimate the effect of the bias.

- Applying these anomalies for hundred years implies that they could/should be part of a global warming projection. Especially in the Pacific, given the shallow mixed-layers, SST-anomalies are largely atmospheric-forced, implying it is unlikely that they are associated with large Q-flux anomalies. Could you show Q-flux anomalies in rcp8.5 projections relative to present-day and/or say something about low-frequency Q-flux variability in CESM. Please put the results and experimental set-up more in context of what we can expect in the future.

- I was missing reference to Graversen et al (Clim Dyn 36, 2103-2112) where it is shown that the strong summer Arctic sea-ice minimum of 2007 can be attributed to anomalous warm winds from the Pacific.

- The bias in Arctic warming and sea-ice decline in in CMIP5 models can only be attributed for a

small part to SST bias further south. In general Arctic sea-ice thickness is too large in the control runs, and for this reason it disappears slower.

- The explanation for the larger role of the Pacific is rather thin and focuses on the Aleutian low, which is not convincing. In both NP and NA warm cases the surface anomalous pressure field features isobars almost perpendicular to the Bering Strait. In the NP-Warm case the response is almost a wavenumber-0 response spinning up the polar vortex and creating a large scale southward expulsion over the American continent into the subtropics. In the NA-Warm experiment the response is more a wavenumber-1 response south of the Arctic and a smaller intensification over the Siberian sector where only cold air can be advected towards the Arctic. Two questions remain unanswered:

- 1. What is the vertical structure of these anomalies? Please inspect 500 and 850 mb as well and provide us with at least 1 extra level and tell in words what the other level is doing.

- The authors implicitly and probably incorrectly assume that the warming is governed by changes in mean flow. I would anticipate the eddy-term dominates here (storms and wave trains), esp. in NP-warm where the polar vortex intensifies. Can they make a rough heat budget analysis integrated over the troposphere and show the anomalous mean and eddy terms?

We would like to thank the reviewers for providing thoughtful and constructive comments on the manuscript “*North Pacific surface ocean heat fluxes enhance Arctic warming*” by S. Praetorius, M. Rugenstein, G. Persad, and K. Caldeira. We have addressed all of the comments and suggestions provided by the reviewers and have made substantial revisions based on their feedback. Changes to the original manuscript include:

- 1) An additional simulation was run to test the impact of an equivalent Atlantic-sized area for the positive heat flux perturbation imposed on the Pacific.
- 2) We have done additional calculations based on the reviewer’s suggestions to provide greater insight into the role of poleward moisture flux in the contribution to Arctic warming.
- 3) We have changed the title to “Global and Arctic climate sensitivity enhanced by changes in North Pacific heat flux” to more clearly articulate the central finding of the paper, as one reviewer suggested.

We have made the following changes to the figures to enhance clarity:

Fig. 1:

- We removed two of the time series on the original figure to help simplify the figure and avoid redundancy.
- We changed the colors of the timeseries so that the associated dots denoting the core locations would be more visible on the map figure 1b.
- We changed the coastlines to white.

Fig. 3&4: We have split these figures into 2 separate figures to accommodate the difference plots of the two simulations, as one reviewer suggested.

Fig. 5: We have updated this figure to reflect the new Moist Static Energy calculations suggested by one of the reviewers.

Fig. S7&8: These figures include the new Q-flux and resulting surface air temperature anomaly for the Atlantic-area equivalent Northeast Pacific positive heat flux simulation.

In addition to the changes made to accommodate the reviewers’ comments (detailed below), we have reconfigured the paper to incorporate more of the supplemental text and figures into the main manuscript, in accordance with the format of Nature Communication articles. We have moved the supplemental Table 1 and the discussion of the possible mechanisms that lead to greater moisture transport into the Arctic in response to the North Pacific heat flux to the main text. We have also included 9 additional references to accommodate reviewers’ suggestions and new additions to the text.

We have included the reviewers’ comments (in italicized text) below, followed with our detailed responses in blue text. Changes made in the manuscript file are highlighted in yellow to make it easy for the reviewers to identify our revisions.

Reviewer #1 (Remarks to the Author):

This manuscript starts from the observation that historical sea surface warming patterns are asymmetric between the N. Pacific and N. Atlantic, with strong warming in the subpolar Northeast Pacific. Various paleo records from the past 18,000 years show that both the N. Atlantic and N. Pacific SSTs increased roughly coevally with temperatures in Greenland during periods of abrupt warming. The authors conduct idealized experiments with a slab ocean configuration of the CESM climate model. The experiments are designed to measure differences in the climatic response to imposed sea surface heat fluxes in the extratropical regions of the two ocean basins. The major finding is that Arctic temperatures are more sensitive to imposed heat flux anomalies in the Pacific than the Atlantic.

The main result is very interesting and worth understanding in detail. Unfortunately I find some serious flaws in this manuscript and I can't recommend it for publication in its current form. My main criticism of the manuscript centers around Figure 4, and the conclusion that differences in moisture flux are the key driver of the differences in Arctic climate response to the Pacific and Atlantic perturbations (e.g. line 20-22).

The conclusion about the moisture flux seems to rely almost entirely on the finding that the zonally integrated moisture flux across 67°N increases more in the NP-WARM case than the NA-WARM case. There are a number of problems with this line of argument, which I outline here.

- First, the presentation is somewhat convoluted and I had a difficult time understanding what the authors were trying to say. For one thing, the numbers in the lower panel of Fig. 4 (+21%, +17%) do not agree with Fig. 4 or Table 1 (+18%, +15%).

We have updated the moisture flux figure following the suggestions below, and we have clarified what the numbers on Fig. 6 represent in the figure legend.

- The discussion on lines 151-153 is confusing. Fig. 4 shows that there is enhanced moisture flux through the Pacific gateway in BOTH cases, and about the same magnitude.

We started from the observation that the North Pacific heat flux perturbations resulted in a greater net moisture anomaly in the Arctic. As the reviewer points out in a later comment, this alone doesn't prove that moisture transport is the primary mechanism, as warmer air will inevitably hold more moisture. The zonal calculations at the boundary of the Arctic (shown in the original Fig. 4) were an attempt at diagnosing whether this greater observed moisture in the Arctic in response to the North Pacific heat flux was a response to a greater poleward moisture flux into the Arctic – and if so – where the majority of the moisture was entering. Perhaps counterintuitively, the North Pacific perturbations showed a greater enhancement of moisture transport through the Atlantic gateway. We did try to articulate this in the original manuscript: “This implies Pacific Ocean warming does not solely lead to enhanced moisture transport directly through the Pacific gateway, but may ultimately be channeled through the Atlantic gateway as well.” However, we have subsequently modified this section in line with the reviewer's suggestions for the MSE calculation below.

These calculations show that the change in northward energy transport is dominated by the change in the latent heat term and that the northward latent heat transport is larger in the NP-Warm case than the NA-Warm case, largely supporting our original inferences about the moisture flux. The total latent heat transport shows a similar pattern to the moisture flux anomaly we originally showed in Fig 4, with a greater increase in latent heat energy transported through the Atlantic gateway in response to the NP-Warm case. However, we know do an additional decomposition (following Reviewer # 3's suggestion) of the total latent heat energy into the mean and eddy terms (Fig. S5), which shows that the mean term largely follows the zonal distribution inferred from the total transport, whereas the eddy term has a more divergent trend in both cases that appears to follow the atmospheric pressure anomaly patterns more closely.

- Line 155: "The relative troposphere-integrated Arctic air temperature changes ... are proportional to the zonal profiles of moisture flux. This supports a dominant role of water vapor and latent heat flux as the primary drivers of the Arctic temperature response"

I had a lot of trouble understanding what the authors meant here. First I assumed that it was a statement about the zonal structures of the temperature and moisture flux anomalies. But looking at the top and bottom panels of Fig. 4, I don't see any such proportionality.

I concluded that the authors were simply referring to the integrated values of Arctic surface temperature and moisture flux listed in Table 1. In this case, yes it is true that both the total moisture flux and Arctic warming are larger in the NP-Warm case.

In any case, this is not a strong argument for causality. All else equal, the moisture flux will increase with warming just due to the increase in saturation, regardless of the mechanism driving the warming.

Apologies for the confusion. We have updated this figure and the accompanying discussion following the suggestions of the reviewer's next comment (detailed below).

- The authors compare sensible heat fluxes and moisture fluxes in a qualitative way.

Unfortunately I think this analysis is incomplete and may be misleading. Fluxes of sensible heat and moisture are both important because they both contribute to the net transport of energy into the polar cap. The relevant metric for atmospheric heat transport is the flux of moist static energy (MSE), which includes the geopotential energy term in addition to sensible and latent heat terms:

$$MSE = cp T + L q + g Z$$

where Z is the geopotential height. The fluxes in Fig. 4 only show the first two terms. On the other hand, analysis such as Overland and Turret (1994) show that the geopotential term is the largest contribution to the MSE flux across 70°N in the lower troposphere.

But even without this information, we can quantitatively compare the relative contributions of sensible and latent heat to the anomalous heating of the Arctic cap by putting both fluxes into common energy units, simply by multiplying sensible heat by $cp = 1004 \text{ J / kg / K}$ (specific heat at constant pressure), and multiplying latent heat by $L = 2.5 \times 10^6 \text{ J / kg}$ (latent heat of vaporization).

From Fig. 4, a typical magnitude of latent heat flux anomaly is about $0.6 \times 10^{-2} \text{ kg/kg m/s}$ --- yielding an anomalous energy flux of $1.5 \times 10^4 \text{ J/kg m/s}$

The zonal average sensible heat flux anomaly is harder to judge eye from Fig. 4, but might be about $0.5 \times 10^3 \text{ K m/s}$ --- yield an anomalous energy flux of $0.5 \times 10^6 \text{ J/kg m/s}$ --- two orders

of magnitude larger than the moisture flux contribution! This doesn't seem plausible but is consistent with the values reported in Figure 4. And it casts serious doubt on the mechanism proposed by the authors. I strongly recommend that the authors do a more careful analysis of the different contributions to the poleward energy flux.

- A related point: I didn't find any information in the methods or supplemental material about how the fluxes were calculated. Are they based on online diagnostics where the correlations (e.g. v^*T , v^*q) are computed at every timestep in the model? Or are they computed offline by multiplying time-averaged v and T fields? Hopefully it is the former, but if the latter, it is critical to document the time frequency of the averaged output that was used to compute the correlations. Also, these are reported as "tropospheric averages". The authors need to be more specific somewhere in the methods about what vertical levels were included in the calculation. Also, these kind of flux diagnostics are notoriously susceptible to spurious mass imbalance issues. The CESM uses a hybrid sigma-pressure vertical coordinate that introduces some complexity to the tropospheric diagnostics, particularly in the vicinity of topography. See appendices in Hill et al. (2015) and Yang et al. (2015) for discussion of the effects of spurious mass imbalance issues on energy flux diagnostics (and how to correct for them) in the GFDL and CESM models respectively.

We thank the reviewer for their helpful suggestion. Following the theory of Yang et al., 2015 [one of the papers cited by the reviewer: Yang et al., Decomposing the Meridional Heat Transport in the Climate System. *Climate Dynamics* 44, no. 9–10 (2015): 2751–68], we calculate the northward total atmospheric energy transport at each latitude, as well as its decomposition into dry static energy (i.e. sensible heat and geopotential energy terms) and moist (i.e. latent heat term) energy. As in Yang et al., 2015, the total atmospheric energy transport through each latitude ($H_a(\Phi)$) is calculated as:

$$H_a(\phi) = 2\pi a^2 \int_{-\pi/2}^{\phi} \cos \phi' (R_{TOA} + F_S) d\phi'$$

for Earth's radius (a), latitude increment (Φ'), top-of-atmosphere radiative flux (R_{TOA}), and surface radiative (F_S), with sign convention such that $R_{TOA} + F_S$ constitutes energy into the atmosphere. The latent heat transport through each latitude ($H_{LH}(\Phi)$) is similarly calculated using the moisture balance equation as

$$H_{LH}(\phi) = 2\pi a^2 \int_{-\pi/2}^{\phi} \cos \phi' L_v (Evap - Precip) d\phi'$$

for latent heat of vaporization (L_v). The dry static energy transport is calculated as the residual, $H_a(\Phi) - H_{LH}(\Phi)$. Each term in the climatology as well as the perturbation in NP-Warm and NA-Warm cases is shown in the figure below (now Fig. 5a).

The calculation shows that the increase in total atmospheric northward energy transport (dotted line) into the Arctic (through 66.6°N) is larger in the NP-Warm Case (blue, 0.054 PW) than in the NA-Warm Case (red, 0.043 PW). The decomposition demonstrates that this is driven by a larger increase in northward latent energy transport (solid line) in the NP-Warm case (blue, 0.081 PW) than in the NA-Warm case (red, 0.068 PW). That is, it demonstrates that larger increased latent energy transport into the Arctic is the driver behind the larger overall increase in total atmospheric energy transport into the Arctic in the NP-Warm case. The latent energy transport increase is partially offset by a decrease in dry static energy transport (dashed line) that is highly similar between the NP-Warm (blue, -0.028 PW) and NA-Warm (red, -0.025 PW) case.

We then analyze where this latent energy is entering the Arctic by analyzing the longitudinal distribution of latent energy transport across 66.6°N (the latitudinal transect in the previous figure – now Fig. 5b, shown below). We calculate this as the longitudinally-resolved latent energy component of the vertically-integrated moist static energy transport:

$$H_{LE} = \int_{p_t}^{p_s} a L_v v q \cos \varphi / g dp$$

integrating between atmospheric pressure at the surface (p_s) and top-of-model (p_t) at latitude ($\varphi=66.6$ N), gravitational acceleration (g) and other variables as in previous equations. The moisture transport (vq) is calculated using the model online diagnostic VQ, which outputs the multiplication of the two terms at each model time step and includes both the mean and the eddy components of the latent energy transport.

Other points:

- We now calculate everything in PW, as the reviewer requests, allowing direct quantitative comparison of the contribution from latent versus dry static energy.
- We now calculate the atmospheric transportation through the full atmosphere, rather than only the troposphere. This prevents issues associated with defining a robust troposphere/stratosphere transition and is a more complete representation of the influence of the perturbations on energy transport into the Arctic (see the appendix of Hill et al. (2015), cited by the reviewer, for further justification of conducting a full-atmosphere calculation).

On the spurious mass balance issue:

- This is absolutely a concern when attempting to use the online diagnostics to close the energy budget. We avoid this by using the atmospheric radiative energy budget ($R_{TOA} + F_S$) and moisture budget (Evap - Precip), respectively, to calculate the total atmospheric energy transport into the Arctic and the contribution from latent energy transport, rather than the online diagnostics VT and VQ.
- The vertical integral of the online diagnostic VQ is used in the longitudinal and eddy vs. mean decompositions (Fig. S5). As in Yang et al. (2015), our vertical integral calculations are on pressure rather than sigma-pressure coordinates. However, because we are only using these calculations to determine the longitudinal gateway through which the latent energy is being most strongly transported, rather than to close the energy budget, the non-conservation issues associated with the sigma-pressure coordinates do not affect the interpretation of the results.

- It's also possible that the energy flux diagnostics are a red herring here, and that the key role of increased moisture fluxes is actually to increase the cloud cover over the Arctic -- making the clouds the "local driver" of the warming rather than a feedback upon warming from enhanced energy transport. Testing this hypothesis would require a set of "locked cloud" simulations, which could shed very interesting light on the mechanisms.

Unfortunately, we cannot do locked-cloud simulations at this time. We agree with the reviewer that the low clouds are likely the primary local driver of Arctic warming. The attempt to uncover the underlying mechanism that led to a greater increase in Arctic low clouds in response to the North Pacific heat flux was what ultimately led us back to the moisture flux analysis. We feel our discussion does implicate low clouds as a primary local driver of Arctic surface temperature change, and hopefully the new calculations and discussion of the latent heat and moisture transport helps to provide the “bridge” to the changes in low cloud cover in the Arctic.

- Regarding the mechanism for the different sensitivities to Pacific vs. Atlantic perturbations: the authors tested the hypothesis about topographic constraints, and found a most null result. I suggest considering a simpler hypothesis about the relative sizes of the Atlantic and Pacific basins. Because the Pacific is wider, the imposed q-flux (Fig. S1) is locally weaker in this case. In a perfectly linear climate system this shouldn't matter since their area-integrated values are identical. However there may be some non-linear effects operating at the sea surface such that the locally weaker Pacific perturbation results in a more "purely evaporative" response. With the clarity of hindsight, it might have been better to set up the experiments with identical q-fluxes in the two basins, with a locally larger Pacific q-flux covering an Atlantic-sized sector on the eastern side of the basin. This would have been a better-controlled experiment, and also quite relevant because of the role of Northeast Pacific SSTs in the observed anomalies in Fig. 1A.

We agree that having a constraint on the impacts of the relative size of the perturbation on the Arctic climate response was a necessary variable to evaluate and have run an additional simulation with an Atlantic-sized positive heat flux perturbation in the Northeast Pacific (NEP-Warm). This simulation results in a qualitatively similar result: the positive heat flux from the Northeast Pacific that is constrained in an Atlantic-area equivalent still results in a greater global and Arctic warming than the positive heat flux from the North Atlantic, and is statistically indistinguishable from the NP-Warm simulation in the global mean surface air temperature response (Table 1, Fig. S9 - below). We have added discussion of this new simulation on lines 210-223.

While we agree that this was an important variable to constrain, we stick to the original heat flux configurations for the primary focus of the manuscript, as this would appear to be a more realistic scenario (i.e. more diffuse heat flux anomalies spread out over a larger area in the North Pacific), especially when considering the artificial truncation of the heat flux anomaly in the central Pacific when forcing the anomaly to be constrained to an Atlantic-sized area (Fig. S7).

Minor comments:

Line 12: shouldn't poleward atmospheric heat transport also be mentioned here?

Yes, we changed this to “oceanic and atmospheric”

Line 50: "ocean SST anomalies of +4°C in the North Pacific" ... This is inconsistent with Fig. 1a, which shows SAT anomalies that are mostly < +2°C

Yes, thank you for catching this. The reference to +4°C was in relation to maximum anomalies observed in the Northeast Pacific during this period (as for example, seen in the St. PAPA SST record plotted on Fig. 1c and referenced in Bond et al., 2015), rather than mean values, which are plotted in Fig. 1a. We have changed this to “paralleling anomalous ocean warming in the Northeast Pacific (16) (Fig. 1a, 1c)”, which includes a callout to both the average values and the yearly means at St. PAPA.

Line 62: The text here should clarify that the "North Atlantic" in this case is a location nowhere near the "warming hole" that is so prominent in Fig. 1

We have changed the SST record plotted on Fig. 1c to a different record that is within the region of the “warming hole”. This core site in the North Atlantic is also at a more similar latitude (56°N) to the SST record from the North Pacific (59°N) that is plotted in Fig. 1c, thus making these two SST records more “apples to apples”. Note, we have also added a reconstruction of broad regional trends in the North Atlantic (for the time period 19-6.5 ka)(Shakun et al., 2012) to help show that the localized change at the specific core site plotted in Fig. 1c is broadly consistent with reconstructed trends from the North Atlantic region at large.

Line 70: The text seems to draw an equivalency between Greenland temperatures and Arctic temperatures as a whole.

While we agree that Greenland is not necessarily reflective of the Arctic as a whole, the ice core records from Greenland are the most high-resolution and well-dated climate archives from the Arctic region that span the deglaciation, thus making these records the ‘gold-standard’ for paleoclimatic records in the Arctic. However, we have added the following sentence and reference (starting on line 64) to broaden the scope of paleoclimate records within the Arctic, so as not to solely use Greenland ice cores as a proxy for Arctic changes as a whole:

“These warming events were also accompanied by enhanced meridional flow of Pacific moisture transport across Alaska (Fischer et al., 2008), implying a strong coupling of extratropical northern hemisphere SST changes with major reorganizations of poleward moisture transport and rapid Arctic climate change.”

The Fischer et al., 2008 reference is an ice core $\delta^{18}\text{O}$ record from Mt. Logan in Alaska that is interpreted to largely reflect changes in moisture transport between zonal and meridional flow. During the two abrupt warming events, this ice core shows heavier $\delta^{18}\text{O}$ values, consistent with an interpretation for stronger meridional flow across Alaska.

Line 81: It's not clear a priori that the important teleconnections linking the midlatitude sea surface to the Arctic will be manifest as changes in meridional heat transport.

Agreed. We changed “meridional heat transport” to “ocean heat flux”.

Line 125: The authors are implicitly assuming that the cloud radiative effect is proportional to changes in cloud amount. However it is important to recognize that this is often not true for the SW effects of low clouds in cold regions. The reason is that the brightness of cold clouds increases strongly with temperature. This is a well-documented physical effect that is also represented in the models, and is a reason why SW cloud feedback tends to be negative at high latitudes (e.g. Zelinka and Hartmann 2012). I'm not sure whether this is an important issue in the present study, but should be acknowledged that cloud amount may be a poor proxy for SW cloud feedback in the Arctic.

We added the following sentences on what is now line 129: “The cloud radiative effect at the top of the atmosphere can still be negative at high latitudes as a result of the short-wave feedback and cloud brightening in response to increasing temperature (Zelinka & Hartmann 2012),

however, we focus here on the net radiative effects at the surface, given the influences on Arctic surface temperature and sea-ice extent.”

Line 135: I appreciate how the authors are carefully avoiding assigning causality here. (This is a compliment, not a criticism)

Thank you.

Line 154: "troposphere-integrated Arctic air temperature" If I understood correctly, what's plotted in Fig. 4 are temperatures at 66°N. The wording seems to imply something like an area-integrated temperature over the Arctic region, which is not the same thing.

We changed “Arctic” to “zonal” to clarify this was the zonal mean not the mean for the entire Arctic.

Line 180: "Changes in moisture transport from the sub-Arctic to the Arctic provide the initial driver for Arctic change..."

See major comments above. This is a reasonably hypothesis but I don't think it has actually been shown here.

We have changed the wording to be less rigid in our interpretation: "Changes in moisture transport from the sub-Arctic to the Arctic *thus plausibly* provide the initial driver for Arctic change..." "

Line 209: "our results challenge the pervasive assumption that abrupt North Pacific warming events in the past were merely a downwind response to North Atlantic ocean circulation changes"

This seems like an overstatement. I don't see anything here to suggest that the hypothetical North Pacific driver would provide a better explanation or fit to the data.

On this point, we respectfully disagree that this inference is not supported by the data and the results of our study. The paleo data indicates that SST changes of similar magnitude occurred in both the North Pacific and North Atlantic. Our simulations are in part an attempt to ‘decouple’ the climate impacts of a North Pacific warming on the order of 4-5°C from the climate impacts of a North Atlantic warming of 4-5°C. Given that both regions are warming with the same magnitude in the past, our simulations suggest that the North Pacific warming could have had a stronger effect on the Arctic than the North Atlantic warming. While we are not trying to dismiss the climate impacts related to changes in circulation and heat transport occurring in the North Atlantic region, most models that attempt to simulate the abrupt warmings in the North Atlantic region do not reproduce a “downwind” warming in the Northeast Pacific anywhere near the magnitude of the SST changes that are documented in the paleo records (e.g. Kageyama et al., 2013). This requires that dynamic changes in the North Pacific are synchronistic with North Atlantic changes, rather than merely a response to them. Our model results (as well as other studies: e.g. Roche et al., 2010, Peteet et al., 1997) also imply that it is “easier” for North Pacific changes to propagate over the North Atlantic region than for North Atlantic changes to affect the Northeastern Pacific.

Additional evidence that corroborates that these warming events involved major Pacific “regime shifts” are the occurrence of widespread ocean hypoxia across the North Pacific margins during these events, indicating *local* oceanographic processes were at play in the North Pacific that affected the entire water column. Furthermore, recent anomalous SST patterns that emerged in the Northeast Pacific (“the blob”) highlight the potential for emergent North Pacific SST variability that is both unexpected and comes with far-reaching climate impacts.

We feel all these separate lines of evidence warrant a reevaluation of the links between North Pacific warming events and rapid Arctic change in the past, rather than resting on the assumption that previous abrupt Arctic climate change events were mostly a function of North Atlantic changes. This is especially relevant for looking towards the future, as it appears more likely that in the near term, there will be stronger warming the North Pacific than in the subpolar North Atlantic (based on model projections).

Line 565-568: These arguments seem inconsistent with the meridional wind and moisture flux anomalies that are actually shown in Fig. 4.

We have reformatted this section in line with the new MSE calculations outlined above, which now eliminates this previous discussion.

References

J. E. Overland and P. Turet. Variability of the atmospheric energy flux across 70°N computed from the GFDL data set. In O. Johannessen, R. Muench, and J. Overland, editors, The Polar Oceans and Their Role in Shaping the Global Environment, number 85 in Geophysical Monographs, pages 313–325. American Geophysical Union, 1994.

S. A. Hill, Y. Ming, and I. M. Held. Mechanisms of forced tropical meridional energy flux change. J. Climate, 28:1725–1742, 2015.

H. Yang, Q. Li, K. Wang, Y. Sun, and D. Sun. Decomposing the meridional heat transport in the climate system. Climate Dynamics, 44:2751–2768, 2015.

M. D. Zelinka and D. L. Hartmann. Climate feedbacks and their implications for poleward energy flux changes in a warming climate. J. Climate, 25:608–624, 2012.

We thank the reviewer for insights from the above references, and have included the Zelinka & Hartmann 2012 reference on line 131, and the Yang et al., 2015 reference is now included in the Methods section.

Reviewer #2 (Remarks to the Author):

The manuscript concerns the effect on the Arctic climate of temperature anomalies in the North Pacific. Based on model experiments it is argued that a heat flux anomaly in the North Pacific has a much larger impact on Arctic climate than has a similar anomaly in the North Atlantic

region. This is a very interesting manuscript that provides a relevant and important contribution to the discussion on Arctic climate change attracting considerable attention these years. The study is comprehensive and creatively executed, and the methods appear robust. I have some minor points and suggestions, and recommend publication of these have been considered.

Points for the authors to consider:

1. The weakest point, in the evidences provided by the author for the importance of the North Pacific heat anomaly, is that the evidences mostly rely on a single climate model, although this model arguable is one of the worlds leading and most state-of-the-art models. I will still recommend to bring forward towards the end of the paper a caution of the results relating on the dynamics of a single climate model.

While we acknowledge the reviewer's concern about our results being demonstrated in only one climate model, we do not have the ability to run additional simulations in other climate models at this time. We note that other studies employing different coupled GCMs show broad-scale similarities to the basic responses we observe of a strong Arctic and northern hemisphere wide climate response to North Pacific cooling (e.g. Peteet et al., 1997, Roche et al., 2010), as well as the more attenuated response of Northeast Pacific SST to changes occurring in the North Atlantic region (e.g. Roche et al., 2010, Okazaki et al., 2010, Kageyama et al., 2013). We hope that our study will encourage more modeling studies to expand upon and further investigate our central findings in a range of climate models and scenarios.

We have added the following starting on line 280:

“Our results are broadly consistent with surface air temperature anomaly patterns generated in a coupled general circulation model in response to forced SST cooling in the North Pacific (Peteet et al., 1997). We note that North Atlantic heat flux anomalies result in relatively weak SST anomalies in the Northeastern sector of the Pacific in our study, consistent with many model studies that exhibit either weak or even reversed temperature anomalies in the Northeastern Pacific in response to forced perturbations in North Atlantic circulation (Okazaki et al., 2010, Kageyama et al., 2013).”

2. Since the fundamental idea is to investigate the effect of change of the ocean circulation, it seems more appropriate to construct all experiments with a zero forcing, as, fundamentally, the ocean circulation redistribute energy but does not produce energy. The sea-saw experiments are constructed that way, but those with forcing in a single ocean are not. It would be more relevant for a comparison across experiments that all experiments have zero forcing. Many earlier work have shown that Arctic amplification appears just as a result of a forcing, but here the intention seem to be to show how Arctic amplification is related to a given ocean heat-flux anomaly. However I am not requiring the authors to redo their experiments, only I am asking the authors to consider what the difference in forcing/no forcing implies for their results in this context.

The central issue with focusing only on the seesaw simulations is that there are non-linearities in the climate response to the positive and negative heat flux anomalies (negative heat fluxes result in a stronger global cooling); thus the redistribution between basins has the confounding element that whatever basin has a negative heat flux will have slightly more leverage on the climate

response. This makes it difficult to separate the regional impacts of the heat flux anomaly (North Pacific versus North Atlantic) from the sign of the heat flux anomaly.

Additionally, when an anomalous heat flux is simultaneously imposed on both basins, it becomes impossible to disentangle the climate effects that are uniquely related to each regional forcing, and it is clear that regional changes in heat flux can have far field effects (e.g. Rose et al. 2014, Rugenstein et al., 2016).

3. L11-12: *Perhaps mention atmospheric circulation change as an additional source of Arctic amplification (e.g. Graversen et al. Nature, 2008).*

We have changed this to “oceanic and atmospheric heat transport” and added the Graversen et al., 2008 reference.

4. L24: *“imply global” -> “imply that global”.*

We have changed this accordingly.

5. L34: *“Thought to be most” -> “often argued to be the most”.*

We have changed this accordingly.

6. L96-99: *Mention here that this sentence refers to experiments with forcing in a single ocean in contrast to the sea-saw experiments discussed just above. In order to see the difference in response to positive and negative forcing more clearly, as discussed in this sentence, difference plots could have been provided in Fig. 3. The sentence in L100-101 could be deleted since this point is likely clear to the reader at this point.*

We added “as can be seen in the single basin positive/negative heat flux simulations (Fig. 3, Fig. 4, Table 1)” to clarify that this statement was in reference to the single basin experiments.

We deleted the sentence in L100-101.

7. L168-169: *The parenthesis can be omitted.*

We removed the parentheses and added a semicolon instead.

8. L169-172: *Perhaps provide a reference to Fig. 4 after this sentence.*

Done.

9. L178: *How does latent heat advection reduces the Planck feedback? Please provide a sentence for explanation.*

We have deleted this sentence, as it wasn't central to the main points of our paper.

10. Fig. 1a and b: Continents are difficult to see. Perhaps provide the coast lines in white on top of the shading.

We have changed the coastlines to white.

11. Fig. 4 caption: Please indicate what the numbers in the lower right corner of the frames mean.

We have changed this figure entirely based on suggestions of Reviewer #1 (now Fig. 5).

12. L461: “that” -> “a”?

We have changed this as suggested.

13. L465: “which follows” -> “with the same pattern as”.

We have changed this as suggested.

14. L466-467: I don't understand this sentence, saying that slab-ocean does not allow lateral transport. As far as I know, the Q-fluxes are indeed the lateral transport.

A Q-flux is the heat flux at the bottom of the mixed layer. This is a three-dimensional surface diagnosed by the fully coupled model with a full depth dynamic ocean. The Q-flux is a two-dimensional field, i.e. including all heat fluxes which leave the mixed layer, be it vertically or horizontally. We generate the Q-flux fields based on the fully coupled model. However, the slab ocean simulation itself does not contain lateral transport within the mixed layer. It is not dynamic. One Q-flux field represents one snap shot in time.

15. L479: Perhaps remove the parenthesis.

Done.

16. L488-490: I Could not follow how this error was estimated. Please provide a sentence or two for clarification.

Errors are calculated as the standard error of the mean multiplied by 1.96, which is the number of standard deviations required to contain 95% of the area of a normal distribution. This is equivalent to the 95% confidence interval of the mean. The calculation of the standard error of the mean assumes a sample size of N independent samples, which in our case are the individual model years. Autocorrelation between model years, however, reduces the number of independent samples, as described by Santer et al., 2000. This autocorrelation can be corrected for by using equation (6) of Santer et al., 2000 to determine an “effective sample size”, N_e , that adjusts the original sample size for this autocorrelation. We use this “effective sample size” (N_e) when calculating our standard error of the mean.

We have elaborated on this on lines 614-621.

17. Fig. S1: Why is different types of map chosen (upper panel shows Pacific in the center, whereas the others show Africa)?

We have changed the control Q-flux to be consistent with the other maps in Fig. S1.

18. Fig. S7: Also here it would be appropriate to show difference plot for the context of the discussion in L541-546.

We have added difference plots for this figure (now Fig. S6).

Reviewer #3 (Remarks to the Author):

In this papers the authors argue that atmospheric energy transport changes are crucial in Arctic warming (which is well known), and that the Pacific plays a major part in this (which is also well known). What is new, however (and should be reflected in the title), is that they demonstrate the Arctic (and global system) appears more sensitive for Pacific heat flux anomalies than for Atlantic ones, even if the global integrated change in ocean heat uptake/release anomaly remains the same. As a result they argue that the Arctic and global climate sensitivity is sensitive to the projected zonal distribution of heat flux anomalies.

Despite a host of caveats, which I will detail below, I think the results, even if the experimental set-up may appear unrealistic, are surprising and insightful, and therefore I recommend publication in Nature Comm after the authors have addressed the comments below.

Following the reviewer's suggestion, we have changed the title to "Global and Arctic climate sensitivity enhanced by changes in North Pacific heat flux" to more clearly articulate the central findings of our study.

- The experimental set-up was unclear to me. First, is the Q-flux as shown in Fig. S1 fixed?

Yes, the Q-fluxes (as shown in Fig. S1) are fixed in time.

If so, the runs are essentially atmosphere only runs and the slab ocean is only used as a diagnostic to derive the Q-flux forcing?

The Q-fluxes account for ocean heat transport divergence. The slab ocean run is a simulation in which the atmosphere has to match the heat flux forcing at the surface. However, the atmosphere is "free" to create the components of the surface energy balance. E.g. locally it has to match 4W/m^2 surface heat flux, but can do so by sensible, latent, long or shortwave as "it pleases", i.e., depending on the atmospheric processes. The simulations are energy conserving, as opposed to fixed-SST simulations, in which the atmosphere "sees" the SST only (and generates its own heat flux accordingly).

If not, please explain in more detail. Also, I assume the Q-fluxes were derived accounting for ocean heat transport divergence. If not the Q-fluxes applied are wrong (biased). Please explain this better and try to estimate the effect of the bias.

The Q-fluxes are derived from a fully coupled (full depth, dynamic) ocean model. We diagnose the mixed-layer depth in this coupled simulation and calculate the heat flux through this surface. This does include ocean heat transport divergence. That field is prescribed to the slab ocean as a Q-flux. The control simulation Q-flux represents reality pretty well (see literature reference in the supplemental material), i.e. the bias is small compared with observations.

- Applying these anomalies for hundred years implies that they could/should be part of a global warming projection. Especially in the Pacific, given the shallow mixed-layers, SST-anomalies are largely atmospheric-forced, implying it is unlikely that they are associated with large Q-flux anomalies. Could you show Q-flux anomalies in rcp8.5 projections relative to present-day and/or say something about low-frequency Q-flux variability in CESM. Please put the results and experimental set-up more in context of what we can expect in the future.

We cannot get the Q-fluxes for RCP8.5, as this is not a standard output in CMIP5 and impossible to create offline for each model. To the atmospheric model it does not make a difference whether the SST pattern are generated by a Q flux forcing or prescribed as SST anomalies. We choose the Q flux forcing approach over the prescribing SSTs because this method conserves energy. Fig. 1 shows the SST anomalies for RCP8.5 simulations are comparable ($\sim 2-4^{\circ}\text{C}$) to the resulting SSTs generated by our Q-flux forcing ($2-5^{\circ}\text{C}$) for the northern hemisphere.

- I was missing reference to Graversen et al (Clim Dyn 36, 2103-2112) where it is shown that the strong summer Arctic sea-ice minimum of 2007 can be attributed to anomalous warm winds from the Pacific.

We have added this reference on line 303-305.

- The bias in Arctic warming and sea-ice decline in in CMIP5 models can only be attributed for a small part to SST bias further south. In general Arctic sea-ice thickness is too large in the control runs, and for this reason it disappears slower.

We have changed this sentence to: “Systematic cold biases in North Pacific and North Atlantic SSTs in CMIP5 models⁴⁸ may thus *partly* lead to an underestimation of Arctic warming and sea ice decline in climate projections...”

- The explanation for the larger role of the Pacific is rather thin and focuses on the Aleutian low, which is not convincing. In both NP and NA warm cases the surface anomalous pressure field features isobars almost perpendicular to the Bering Strait. In the NP-Warm case the response is almost a wavenumber-0 response spinning up the polar vortex and creating a large scale southward expulsion over the American continent into the subtropics. In the NA-Warm experiment the response is more a wavenumber-1 response south of the Arctic and a smaller intensification over the Siberian sector where only cold air can be advected towards the Arctic. Two questions remain unanswered:

- 1. What is the vertical structure of these anomalies? Please inspect 500 and 850 mb as well and provide us with at least 1 extra level and tell I words what the other level is doing.

We have added the 500 mb contours to the plots of sea-level pressure in Fig. S10. Both the 500 and 850 mb levels follow the overall pattern apparent in the sea-level pressure anomaly pattern.

- The authors implicitly and probably incorrectly assume that the warming is governed by changes in mean flow. I would anticipate the eddy-term dominates here (storms and wave trains), esp. in NP-warm where the polar vortex intensifies. Can they make a rough heat budget analysis integrated over the troposphere and show the anomalous mean and eddy terms?

We now do a heat budget analysis in response to Reviewer #1's comments, which shows the role of total (mean and eddy) latent energy transport in the Arctic warming. The analysis of the longitudinal structure is conducted using the model diagnostic northward moisture transport (VQ), which calculates the multiplication at every model time step, and thereby includes both the mean and the eddy transport.

Although we did not make any statements in the original text about whether the transport was dominated by mean or eddy components, we do appreciate the reviewer's suggestion to deconvolve these terms. We have now added a figure that shows the all-atmosphere-integrated into the mean and eddy components in the NP-Warm case and NA-Warm case to address the reviewer's concern (Fig. S5, shown below). These show that the longitudinal structure of the change in total transport follows the longitudinal structure of the change in mean transport. However, the reviewer is correct that the eddy transport increases at most longitudes in both cases and drives the total latent energy transport increase in both cases. The longitudinal integral of each term indicates that the mean and eddy changes contribute 22.9% and 77.1% of the total latent energy transport increase in the NA-Warm Case and 23.6% and 76.4% in the NP-Warm case.

REVIEWERS' COMMENTS:

Reviewer #1 (Remarks to the Author):

Review of "Global and Arctic climate sensitivity enhanced by changes in North Pacific heat flux" by S. Praetorius, M. Rugenstein, G. Persad, and K. Caldeira.

This is revised version of a manuscript I reviewed some months ago. Overall I think the authors have done a great job responding to my concerns and those of the other reviewers. This manuscript is clearer and more compelling. The major change in response to my comments was a more consistent and in-depth analysis of the energy transport anomalies. It looks to me like the analysis was done well and presented clearly, and it definitely helps make the case that atmospheric moisture fluxes across 66.6N are implicated in the enhanced sensitivity to Pacific vs. Atlantic Ocean heat fluxes.

Despite the much better clarity of this version, I am still confused about the main physical argument regarding the causal role of that enhanced moisture transport. Increased moisture flux into the Arctic could lead to warming through any of at least three mechanisms:

- (1) the direct warming effect of condensation (i.e., the convergence of the latent heat transport)
- (2) Moistening the column, giving a stronger clear-sky water vapor greenhouse effect
- (3) Moistening and condensation creating more low clouds, which (at least in a strongly temperature-stratified atmosphere) give a stronger greenhouse effect

Some of this is laid out in lines 180-185 but can be clarified further. In reality (and in the model) these 3 effects are difficult to disentangle. Effect (3) is dealt with most clearly in the manuscript, and the schematic in Fig. 6 emphasizes effect (3). Is this focus on effect (3) justified?

The magnitude of effect (1) can be estimated by converting the energy transport anomaly numbers from the manuscript into area-averaged convergence over the polar cap. I looked at these numbers. With the boundary at 66.6N the polar cap area is $2.1 \times 10^{13} \text{ m}^2$. Convergence is just the transport divided by this number, so that 1 PW = 47 W/m² convergence. Then for example the 0.081 PW anomaly in latent heat transport for the NP-Warm case is equivalent to a direct heating of 3.9 W/m², but this is offset by a decrease in DSE transport of 0.028 PW = 1.3 W/m². So the net heating is 2.5 W/m². This is very comparable to the increased LW cloud radiative effect numbers quoted in the manuscript, which suggests that effect (1) might be equally important to effect (3). I would appreciate seeing these numbers compared directly in the paper.

Separating out effects (2) and (3) could be done with radiative kernel analysis, e.g. as done for the polar cap by Singh et al. (2017). As I mentioned previously, "locked cloud" simulations would also be very insightful here, as they would reveal aspects of the response governed more by effects (1) and (2).

I don't necessarily think that the causality of the complex problem needs to be worked out in detail in order to justifying publishing these very interesting results. But unjustified confidence in a causal mechanism that hasn't been clearly shown should be avoided. A clearer discussion of different possible mechanisms along the lines I laid out above would be very helpful.

Line-by-line comments:

Line 130: I think you mean here the *change* in cloud radiative effect could be negative

Line 185: "latent heat fluxes increase" here I think you're talking about local evaporation rather than transport of water vapor into the Arctic. Maybe just say "evaporation increases" instead for

clarity

Line 214: It seems strange, given the effort the authors have gone to here to address the concern about different ocean basin size, that they wouldn't run it out for another 40 years. It's a small additional amount of simulation relative to the runs already reported, and it would clear up this big caveat associated with interpreting the meaning of this result. Unfortunately I'm not too sure whether I should see the reduced response relative to the whole-Pacific perturbation as meaningful or not. If there is to be another round of revision of this manuscript, perhaps the authors will have time to do the additional simulation and this problem will go away.

Line 228: This section is less convincing than other parts of the manuscript, as outlined nicely by Reviewer #3. In particular "signaling an overall increase in the Aleutian low" is problematic. The anomaly is positive over the Aleutian region in this case too. Maybe better to describe it as a shift of the low to the east in this case?

Line 266: "that extend over a greater surface area". I'm not sure why the authors emphasize this difference here? They have already concluded that differences in surface area are not important

Figure 1: This is a beautiful figure. Panel C is outstanding. Well done.

Line 574: Somewhere it needs to be stated that CAM4 is an out-of-date model and there have been significant improvements to representations of low clouds and shallow convection in CAM5 and beyond! This could be key since low cloud changes are at the heart of the mechanism studied here. The fact that these simulations use an old version of the model is buried in the methods section and might easily be missed by most readers. This confusion shows up in Reviewer #2's comment that the CESM "is one of the worlds leading and most state-of-the-art models". That statement just does not apply to the atmospheric component of the model used in this study.

Figure S4: I assumed that the low cloud changes are area-averaged over the whole Arctic polar cap, but this isn't actually stated anywhere as far as I could see.

References

H. A. Singh, P. J. Rasch, and B. E. J. Rose. Increased ocean heat convergence into the high latitudes with co2-doubling enhances polar-amplified warming. *Geophys. Res. Lett.*, 44, 2017.

Reviewer #2 (Remarks to the Author):

Second peer review of "Global and Arctic climate sensitivity enhanced by changes in North Pacific heat flux" by Praetorius et al.

This is a very interesting work indicating that mid-latitude ocean anomalies have wide-spread impact on climate, especially in the Arctic, through the latent heat transport. The authors have done a great job to improve the manuscript relative to the previous version and provide thorough and reasonable answers to our concerns of the previous review round. I suggest publishing of the manuscript – almost – in the current form. I found some minor unclarity for Fig. 5:

1. Line 21-33: "up to" -> "by", remove "more than".

2. Fig. 5: Panel A: I assume the transports from the forcing runs are anomalies and not absolute values, which should be indicated. Panel B: Again, are these absolute transports or anomalies? Here I have the feeling that the forcing results are absolute values given their resemblance to the

control. Units of this panel should likely be in PW/m (Watt per meter). The associated text (Lines 171-179) claims that this figure shows a decomposition into mean and eddy part, but I couldn't find that from the figure. Note that from the Reynolds averaging (that I assume is referred to here) the mean part is based on $\text{mean}(v) \cdot \text{Mean}(E)$, whereas what is shown in the inset of Panel B is $\text{mean}(vE)$, if I understood it correctly.

Good work!

Reviewer #3 (Remarks to the Author):

The authors have provided an extensive and very detailed response to the reviewers comments and have made a substantial revision in accordance with those comments. I recommend to accept for publication pending one minor remark.

In one remark I suggest to compare the flux anomalies in NP-warm and NA-warm with flux anomalies in CMIP5. I may have been unclear in my suggestion. What I ask is to calculate the anomalous surface heat flux in the RCP8.5 projection from either the full CMIP5 MME (or a subset of those models outputting the surface heat flux), or, just from CESM. I anticipate that the differences between those anomalies in NP and NA are much smaller than the 60 W/M^2 anomalies imposed in the experiments discussed here. The reason is that the coupled system adjusts to such heat uptake/release anomalies (finite heat capacity), while in the forced experiments the ocean keeps forcing with the same Q-anomaly, irrespective of the atmospheric response to it. I do not want to imply that this disqualifies the experimental set-up, but do suggest the results are put in proper context by comparing the imposed forcing with what we expect from an RCP8.5 scenario.

Author response to Reviewers' comments for second revision: "Global and Arctic climate sensitivity enhanced by changes in North Pacific heat flux" by Praetorius et al.

Below the reviewers' comments are listed in black, followed by the author's response in blue text.

Reviewer #1 (Remarks to the Author):

Review of "Global and Arctic climate sensitivity enhanced by changes in North Pacific heat flux" by S. Praetorius, M. Rugenstein, G. Persad, and K. Caldeira.

This is revised version of a manuscript I reviewed some months ago. Overall I think the authors have done a great job responding to my concerns and those of the other reviewers. This manuscript is clearer and more compelling. The major change in response to my comments was a more consistent and in-depth analysis of the energy transport anomalies. It looks to me like the analysis was done well and presented clearly, and it definitely helps make the case that atmospheric moisture fluxes across 66.6N are implicated in the enhanced sensitivity to Pacific vs. Atlantic Ocean heat fluxes.

Despite the much better clarity of this version, I am still confused about the main physical argument regarding the causal role of that enhanced moisture transport. Increased moisture flux into the Arctic could lead to warming through any of at least three mechanisms:

- (1) the direct warming effect of condensation (i.e., the convergence of the latent heat transport)
- (2) Moistening the column, giving a stronger clear-sky water vapor greenhouse effect
- (3) Moistening and condensation creating more low clouds, which (at least in a strongly temperature-stratified atmosphere) give a stronger greenhouse effect

Some of this is laid out in lines 180-185 but can be clarified further. In reality (and in the model) these 3 effects are difficult to disentangle. Effect (3) is dealt with most clearly in the manuscript, and the schematic in Fig. 6 emphasizes effect (3). Is this focus on effect (3) justified?

The magnitude of effect (1) can be estimated by converting the energy transport anomaly numbers from the manuscript into area-averaged convergence over the polar cap. I looked at these numbers. With the boundary at 66.6N the polar cap area is $2.1 \times 10^{13} \text{ m}^2$. Convergence is just the transport divided by this number, so that 1 PW = 47 W/m² convergence. Then for example the 0.081 PW anomaly in latent heat transport for the NP-Warm case is equivalent to a direct heating of 3.9 W/m², but this is offset by a decrease in DSE transport of 0.028 PW = 1.3 W/m². So the net heating is 2.5 W/m². This is very comparable to the increased LW cloud radiative effect numbers quoted in the manuscript, which suggests that effect (1) might be equally important to effect (3). I would appreciate seeing these numbers compared directly in the paper.

Separating out effects (2) and (3) could be done with radiative kernel analysis, e.g. as done for the polar cap by Singh et al. (2017). As I mentioned previously, "locked cloud" simulations would also be very insightful here, as they would reveal aspects of the response governed more by effects (1) and (2).

I don't necessarily think that the causality of the complex problem needs to be worked out in detail in order to justifying publishing these very interesting results. But unjustified confidence in a causal mechanism that hasn't been clearly shown should be avoided. A clearer discussion of different possible mechanisms along the lines I laid out above would be very helpful.

Thank you for the suggestions. We have done the suggested calculations and provided additional discussion of the various pathways that increased moisture flux to the Arctic could enhance warming. The additions to the text are highlighted below.

Starting on line 162:

“Within the Arctic, the NP-Warm simulation increases the low cloud formation more (+12.4±0.89%) than the NA-Warm simulation (+9.5±0.87%). The effect of low clouds accounts for an additional 2.9±0.22 W/m² of surface longwave radiative forcing within the Arctic for the NP-Warm simulation versus an additional 2.2±0.22 W/m² in the NA-warm simulation (calculated as the difference in downwelling longwave radiation at the surface between all-sky and clear-sky conditions). Clear-sky downwelling longwave radiation also increases more strongly in the NP-Warm case (10.7±0.71 W/m²) than in the NA-Warm case (7.3±0.69 W/m²). The enhanced longwave radiation contributes to surface warming and sea-ice retreat, increasing the net shortwave radiation absorbed at the sea surface through the reduction in surface albedo (+1.7±0.42 W/m² and 0.9±0.37 W/m² for the NP-Warm and NA-Warm simulations, respectively).”

Starting on line 175:

“Decompositions of the total northward atmospheric energy transport into the dry static energy (sensible heat and geopotential energy terms) and moist (latent heat term) energy terms indicate that the total atmospheric northward energy transport into the Arctic is larger in the NP-Warm case (0.054 PW) than in the NA-Warm Case (0.043 PW), and that this difference is primarily driven by a larger increase in northward latent energy transport in the NP-Warm case (0.081 PW) than in the NA-Warm case (0.068 PW) (Fig. 5a). The increase in northward latent energy transport is partially offset by a decrease in dry static energy transport that is similar between the NP-Warm (-0.028 PW) and NA-Warm (-0.025 PW) cases. This translates to a total of 2.6 W/m² of direct Arctic heating from atmospheric energy convergence in the NP-Warm case and 2.1 W/m² in the NA-Warm case. Analysis of the longitudinal distribution of the northward latent energy transport along the Arctic boundary indicates the dominant regions that this latent energy is entering the Arctic in each case (Fig. 5b). In both the NA-Warm and NP-Warm cases, the total latent heat transport (i.e. amount of moisture intrusion) is enhanced across the Pacific and Atlantic basins relative to the control. The NP-Warm simulation shows a greater total latent heat transfer across the Arctic boundary through the Atlantic gateway, whereas the NA-Warm simulation shows greater total latent heat transfer over Siberia, relative to the NP-Warm case.”

Starting on line 202:

“The larger net moisture transport into the Arctic in the NP-Warm case results in enhanced Arctic surface warming through a number of processes and feedbacks: greater latent heat transfer from the subpolar extratropics to Arctic, greater increase in the area of low clouds in the Arctic and attendant absorption of infrared radiation, and greater amounts of atmospheric water vapor in the Arctic and attendant clear-sky absorption of infrared radiation. The relative magnitude of these

effects discussed above (2.6 W/m² of increased total atmospheric energy convergence, 2.9 W/m² of increased cloud-sky surface longwave radiation, 10.7 W/m² of increased clear-sky surface longwave radiation, respectively) suggests that all three mechanisms play an important role in Arctic warming. There is additionally greater sea-ice decline, which increases surface radiation absorption through reduction in surface albedo (ice-albedo feedback) (Fig. 6). As sea ice retreats, evaporation increases, driving additional low cloud formation and atmospheric moistening. Changes in moisture transport from the sub-Arctic to the Arctic thus plausibly provide the initial driver for Arctic change and these changes are then amplified by local processes.”

We also added the following in the methods section (starting line 454):

“The direct Arctic heating from atmospheric energy convergence is calculated as the total atmospheric northward energy transport values divided by the area of the Arctic (2.089x10¹³ m²).”

Line-by-line comments:

Line 130: I think you mean here the *change* in cloud radiative effect could be negative

Thank you. We have added “The net change in the cloud radiative effect at the top of the atmosphere...”

Line 185: “latent heat fluxes increase” here I think you're talking about local evaporation rather than transport of water vapor into the Arctic. Maybe just say "evaporation increases" instead for clarity

We have replaced “latent heat fluxes” with “evaporation increases”

Line 214: It seems strange, given the effort the authors have gone to here to address the concern about different ocean basin size, that they wouldn't run it out for another 40 years. It's a small additional amount of simulation relative to the runs already reported, and it would clear up this big caveat associated with interpreting the meaning of this result. Unfortunately I'm not too sure whether I should see the reduced response relative to the whole-Pacific perturbation as meaningful or not. If there is to be another round of revision of this manuscript, perhaps the authors will have time to do the additional simulation and this problem will go away.

Unfortunately, we are not able to run the simulation out for any longer at this point.

Line 228: This section is less convincing than other parts of the manuscript, as outlined nicely by Reviewer #3. In particular “signaling an overall increase in the Aleutian low” is problematic. The anomaly is positive over the Aleutian region in this case too. Maybe better to describe it as a shift of the low to the east in this case?

We have changed the wording of this section so as not to overly attribute the surface pressure changes in the Pacific case to patterns associated with the Aleutian low (whereas in the resulting Atlantic anomaly, it does appear more consistent with a weakening of the Aleutian low).

Starting on line 253:

“Sea-level pressure changes in the NA-Warm simulation produce a high-pressure anomaly over the North Pacific that is most pronounced in winter (Supplementary Figure S10), indicating a general weakening of the Aleutian Low in response to a positive North Atlantic heat flux. A stronger Aleutian Low is associated with the transport of warm moist air from the North Pacific into the Arctic through the Bering Strait³⁶, whereas a weakened Aleutian Low shifts the North American ridge into the central Pacific, where it effectively blocks west-to-east propagating storm systems that carry moisture northward³⁷. In contrast, the NP-Warm simulation produces a low-pressure anomaly that extends from the Northeastern Pacific into the Arctic during winter.”

Line 266: “that extend over a greater surface area”. I’m not sure why the authors emphasize this difference here? They have already concluded that differences in surface area are not important

We have changed this sentence to remove the reference to the area: “Our simulations suggest that more diffusely distributed ocean heat fluxes in the North Pacific may have the capacity to exert stronger influences on global and Arctic climate than more concentrated heat fluxes in the North Atlantic through modulations in low cloud cover and poleward moisture transport.”

Figure 1: This is a beautiful figure. Panel C is outstanding. Well done.

Thank you.

Line 574: Somewhere it needs to be stated that CAM4 is an out-of-date model and there have been significant improvements to representations of low clouds and shallow convection in CAM5 and beyond! This could be key since low cloud changes are at the heart of the mechanism studied here. The fact that these simulations use an old version of the model is buried in the methods section and might easily be missed by most readers. This confusion shows up in Reviewer #2’s comment that the CESM “is one of the worlds leading and most state-of-the-art models”. That statement just does not apply to the atmospheric component of the model used in this study.

We state in the introduction the version of the model we use (line 88).

We have also added a mention to this caveat to the sentence starting on line 273:

“Some of the asymmetry in the Arctic climate response to North Pacific or North Atlantic heat fluxes in our simulations may in part be related to model biases or poor representations of cloud physics and shallow convective processes, such as the overly strong Pacific center of the wintertime Arctic Oscillation observed in many climate models³⁸, or the tendency for models to underestimate moisture intrusions to the Arctic through the North Atlantic gateway and overestimate moisture intrusions from the Pacific sector³⁹.”

Figure S4: I assumed that the low cloud changes are area-averaged over the whole Arctic polar cap, but this isn’t actually stated anywhere as far as I could see.

We have clarified that these calculations are for the area-averaged Arctic region in the figure caption.

References

H. A. Singh, P. J. Rasch, and B. E. J. Rose. Increased ocean heat convergence into the high latitudes with co2-doubling enhances polar-amplified warming. *Geophys. Res. Lett.*, 44, 2017.

Reviewer #2 (Remarks to the Author):

Second peer review of “Global and Arctic climate sensitivity enhanced by changes in North Pacific heat flux” by Praetorius et al.

This is a very interesting work indicating that mid-latitude ocean anomalies have wide-spread impact on climate, especially in the Arctic, through the latent heat transport. The authors have done a great job to improve the manuscript relative to the previous version and provide thorough and reasonable answers to our concerns of the previous review round. I suggest publishing of the manuscript – almost – in the current form. I found some minor unclarity for Fig. 5:

1. Line 21-33: “up to” -> “by”, remove “more than”.

Changed as suggested.

2. Fig. 5: Panel A: I assume the transports from the forcing runs are anomalies and not absolute values, which should be indicated. Panel B: Again, are these absolute transports or anomalies? Here I have the feeling that the forcing results are absolute values given their resemblance to the control. Units of this panel should likely be in PW/m (Watt per meter). The associated text (Lines 171-179) claims that this figure shows a decomposition into mean and eddy part, but I couldn't find that from the figure. Note that from the Reynolds averaging (that I assume is referred to here) the mean part is based on $\text{mean}(v) \cdot \text{Mean}(E)$, whereas what is shown in the inset of Panel B is $\text{mean}(vE)$, if I understood it correctly.

Yes, the control lines and axes on both panels are absolute values, whereas the blue and red NP-/NA-Warm lines and the left axis are the delta value (the difference between NP-warm or NA-warm and the control). We have clarified this on the figure caption.

The reference to the decomposition into mean and eddy components was in reference to Supplementary Figure 5 (as opposed to Fig. 5). Now the callouts to the supplementary figures are more explicitly written out, whereas previously they were just listed as Fig. S5, which was easy to miss.

Reviewer #3 (Remarks to the Author):

The authors have provided an extensive and very detailed response to the reviewers comments and have made a substantial revision in accordance with those comments. I recommend to accept for publication pending one minor remark.

In one remark I suggest to compare the flux anomalies in NP-warm and NA-warm with flux anomalies in CMIP5. I may have been unclear in my suggestion. What I ask is to calculate the anomalous surface heat flux in the RCP8.5 projection from either the full CMIP5 MME (or a subset of those models outputting the surface heat flux), or, just from CESM. I anticipate that the differences between those anomalies in NP and NA are much smaller than the 60 W/M^2 anomalies imposed in the experiments discussed here. The reason is that the coupled system adjusts to such heat uptake/release anomalies (finite heat capacity), while in the forced experiments the ocean keeps forcing with the same Q-anomaly, irrespective of the atmospheric response to it. I do not want to imply that this disqualifies the experimental set-up, but do suggest the results are put in proper context by comparing the imposed forcing with what we expect from an RCP8.5 scenario.

We have calculated the RCP8.5 surface heat flux anomalies relative to the preindustrial control in 37 CMIP5 models (Supplementary Figure 12). The annual average gross heat fluxes are the same order as our heat flux perturbations, so our scenario would be the right order of magnitude for major changes in ocean circulation and heat flux patterns in these basins, such as a switching of deep water formation from the North Atlantic to the North Pacific, as has been implicated in the past. However, our test perturbation fluxes are nearly (but not quite) an order of magnitude greater than average perturbations projected for RCP8.5 (with the exception of the North Atlantic region), so in this context our simulations are relevant for understanding how effects might scale with the magnitude of heat flux changes (or possibly shorter-term anomalies in surface heat flux), rather than being directly applicable to future projections.